# Nucleolar- and Nuclear-Stress-Induced Membrane-Less Organelles: A Proteome Analysis through the Prism of Liquid–Liquid Phase Separation

**DOI:** 10.3390/ijms241311007

**Published:** 2023-07-02

**Authors:** Yakov I. Mokin, Anastasia A. Gavrilova, Anna S. Fefilova, Irina M. Kuznetsova, Konstantin K. Turoverov, Vladimir N. Uversky, Alexander V. Fonin

**Affiliations:** 1Laboratory of Structural Dynamics, Stability and Folding of Proteins, Institute of Cytology, Russian Academy of Sciences, St. Petersburg 194064, Russia; mokinyakov@mail.ru (Y.I.M.); asultanbekova@incras.ru (A.A.G.); anny.fefilova@gmail.com (A.S.F.); imk@incras.ru (I.M.K.); kkt@incras.ru (K.K.T.); 2Department of Molecular Medicine, Morsani College of Medicine, University of South Florida, Tampa, FL 33612, USA; 3USF Health Byrd Alzheimer’s Research Institute, Morsani College of Medicine, University of South Florida, Tampa, FL 33612, USA

**Keywords:** amyloid bodies, intrinsically disordered protein, liquid–liquid phase separation, membrane-less organelle, nuclear stress bodies, nucleolar biomolecular condensates

## Abstract

Radical changes in the idea of the organization of intracellular space that occurred in the early 2010s made it possible to consider the formation and functioning of so-called membrane-less organelles (MLOs) based on a single physical principle: the liquid–liquid phase separation (LLPS) of biopolymers. Weak non-specific inter- and intramolecular interactions of disordered polymers, primarily intrinsically disordered proteins, and RNA, play a central role in the initiation and regulation of these processes. On the other hand, in some cases, the “maturation” of MLOs can be accompanied by a “liquid–gel” phase transition, where other types of interactions can play a significant role in the reorganization of their structure. In this work, we conducted a bioinformatics analysis of the propensity of the proteomes of two membrane-less organelles, formed in response to stress in the same compartment, for spontaneous phase separation and examined their intrinsic disorder predispositions. These MLOs, amyloid bodies (A-bodies) formed in the response to acidosis and heat shock and nuclear stress bodies (nSBs), are characterized by a partially overlapping composition, but show different functional activities and morphologies. We show that the proteomes of these biocondensates are differently enriched in proteins, and many have high potential for spontaneous LLPS that correlates with the different morphology and function of these organelles. The results of these analyses allowed us to evaluate the role of weak interactions in the formation and functioning of these important organelles.

## 1. Introduction

In the 21st century, there have been at least two revolutionary changes in the understanding of the structural and functional organization of proteins [1]. At the turn of the century, several groups almost simultaneously came to the conclusion that there are functionally active proteins that do not have an ordered structure. In fact, the structure of an intrinsically disordered protein (IDP) or an intrinsically disordered protein region (IDPR) can be described as a highly dynamic, conformational ensemble representing a dynamic equilibrium between different conformers separated by negligible energy barriers. As a result, any slight change in external conditions (e.g., temperature, ionic strength of the solution, interaction with a partner, post-translational modifications) can lead to a significant change in the energy surface of the protein. This ensures the multifunctionality of IDPs/IDPRs. The degree and penetrance of disorder in a protein can be different, where the proteins can be disordered to different degrees. Some proteins are disordered as a whole, whereas others are known as hybrid proteins, as they contain ordered regions/domains and IDPRs that can also be differently disordered [2,3].

IDPs are commonly multifunctional proteins that are involved in a large number of low affinity and high specificity interactions, which require more structural flexibility [4,5]. As the proteomes become more complex, their proportions of proteins with functional IDPRs increase. The sequences of IDPs are of low complexity and often contain numerous charged and polar residues, being depleted in hydrophobic residues [2].

Low energy barriers between different conformers make it possible for the same polypeptide chain to have a different structure at different times. The stochasticity of IDPs determines the probabilistic nature of their interaction with partners. Thus, according to the new model of transcription activation proposed by Alexander Erkine, the interaction of disordered transcription activation domains with promoters, which causes chromatin rearrangement, has a pronounced stochastic character [6].

Ground-breaking changes in the understanding of the complex relationships between the structure and function of proteins were followed by ground-breaking changes in the understanding of the spatiotemporal organization of intracellular space [1]. The paradigm of the rigid compartmentalization of the intracellular space into a series of well-recognized membrane-bound compartments was replaced by the understanding that a significant part of cellular processes is determined by the functioning of liquid-droplet-like condensates—membrane-less organelles (MLOs) [7]. The formation of MLOs is linked to the property of polymers to separate into phases under the conditions of macromolecular crowding, and this process can be controlled by various external factors and environmental changes (changes in temperature, pH, ionic strength of the solution, and a number of other factors) [8]. These structures can be considered as a metastable condensed state of the intracellular matter.

IDPs and IDPRs play a decisive role in liquid–liquid phase separation (LLPS), leading to the formation of MLOs. As a rule, IDPs/IDPRs contain blocks of the same type of amino acid residues, multiple weak interactions between which (electrostatic, π-π, and cation-π) often initiate the transition of IDPs to the liquid-droplet phase [4,8]. The absence of a membrane determines the dynamics of MLOs and their composition; however, under certain conditions, MLOs are able to form stable hydrogels and even insoluble aggregates. In particular, a number of neurodegenerative diseases are associated with the formation of stress granules by mutant proteins and the formation of amyloid fibrils in them [9].

Weak non-specific inter- and intramolecular interactions of disordered polymers, primarily IDPs and RNA, play a central role in initiating the formation of MLOs and in the regulation of their composition and functional activity. In some cases, the “maturation” process of MLOs is accompanied by a “liquid–gel” transition of their structure, where other types of interactions can play a significant role [10,11].

In this work, we analyzed the propensity for spontaneous phase separation of the proteomes of two MLOs formed in the nucleus in response to stress, which have a partially overlapping composition, but a different functional activity and morphology: A-bodies (amyloid bodies) and nuclear stress bodies [12,13].

In response to various types of stress, the formation of RNA–protein condensates containing non-coding RNA (rIGSRNA) from the intergenic spacer region (IGS) of the rDNA cassette is induced in the cells of some eukaryotes [14]. Initially, these condensates have liquid-droplet characteristics, but subsequently transform into amyloid-like structures, the so-called A-bodies, containing a number of proteins in the fibrillar state [15,16,17]. The marker proteins of these structures are von Hippel–Lindau tumor suppressor protein (VHL) and Cell division cycle 73 (CDC 73) [18].

It is assumed that rIGSRNAs (rIGSRNA16, rIGSRNA22—heat shock, rIGSRNA28—acidosis) expressed in response to a specific stress interact with positively charged regions of some proteins, in particular VHL, resulting in the formation of condensates, which eventually transform into aggregates of amyloid fibrils [18,19]. rIGSRNAs are likely capable of spontaneous phase separation with the subsequent recruitment of proteins to the already pre-formed condensate [20,21]. Although A-bodies have a fibrillar structure, they are physiological and perform important biological functions. A-bodies include biogenesis and RNA processing factors and transcription factors as well as proteins involved in the regulation of metabolism and cell cycle and nuclear translation [22]. The A-body’s proteome significantly depends on the type of stress. It has been shown that only ~20% of proteins in the proteomes of an A-body formed in response to heat shock and acidosis overlap [18].

In response to various types of stress, in the nuclei of primate cells, another type of membrane-less organelle is formed: nuclear stress bodies (nSBs) [23]. The assembly of nSBs is initiated by the activation of pericentric heterochromatin at the 9q12 locus under the control of heat shock transcription factor 1 (HSF1) and leads to the transcription of long non-coding RNA (lncRNA) from the pericentromeric repeat arrays of HSATIII chromosome 9. It is believed that such transcripts are the foci of nSB formation [24,25,26]. These organelles play an important role in the regulation of gene expression on a genome-wide scale due to the massive recruitment of histone acetylases (HATs) and transcription factors to nSBs [27], and in promoting the repression of pre-mRNA splicing [28,29,30]. nSBs are dynamic liquid-droplet condensates; however, with an increase in the duration of stress exposure, these organelles gradually harden and can even form insoluble structures [23]. It is known that a large number of proteins are involved in the formation and functioning of nSBs. Approximately 140 proteins linked to nSBs have been identified, many of which are RBPs (RNA-binding proteins) and are involved in mRNA splicing, processing, and export [28].

Therefore, A-bodies and nSBs are largely similar organelles with an initially liquid-like structure, formed in response to the same stimulus with the participation of non-coding RNA and having overlapping functions (RNA processing). It is believed that the initial stage of formation of both organelles is the accumulation of non-coding transcripts. However, the VHL A-body marker protein translocates into the nucleus in response to stress without interacting with RNA [31], and the HSF1 nSB marker protein is capable of RNA-free LLPS under stress conditions. In this work, we performed a comparative analysis of the proteomes of these organelles for the propensity of their constituent proteins to phase separate spontaneously, undergo phase separation induced by interactions with partners, or to aggregate and form amyloid-like fibrils. This made it possible to obtain indirect data on the mechanisms of formation of these organelles.

## 2. Results

### 2.1. Study Design

First, we collected proteomes of acidosis-induced A-bodies, heat-shock-induced A-bodies, and heat-shock-induced nuclear stress bodies (nBs). A-body proteomes were characterized by one group using mass-spectrometric analysis of amino acid stable isotopes labeled MCF-7 cells (SILAC-MS) before and after appropriate stress [18,19]. According to these works, proteins whose relative content in the A-bodies was two-fold higher than their relative content in the untreated cells were considered as proteins included into the A-bodies. The proteome of heat-shock-induced nuclear stress bodies was obtained by another team using chromatin isolation for the comprehensive identification of RNA-binding proteins by mass spectrometry (ChIRP-MS) and an antisense oligonucleotide (ASO) of HSATIII lncRNA as a stable core component of nSBs formed after the heat shock of HeLa cells [28]. Of course, different methods for obtaining the proteomes of A-bodies and nSBs may have an effect on our analysis. Furthermore, one should also keep in mind that given the difficulties associated with isolating membrane-less organelles (loss of proteins, contamination with components that do not reside in the compartment), conclusions based on proteomics data are inherently prone to error. 

We analyzed the amino acid sequences of collected proteins for the potential propensity to spontaneous and induced liquid–liquid phase separation (LLPS) and the tendency of these proteins to gelation and amyloid fibrillation. We estimated the following values, summarized in Figure 1:-Intrinsic disorder using D^2^P^2^ and RIDAO services;-Tendency to spontaneous LLPS using FuzDrop and PSPredictor services;-Tendency to inducible LLPS using FuzDrop service;-Charge and hydrophobic properties;-Tendency to amyloid fibrillation using AggreScan service;-UniProt and CPAD database search amyloid-forming proteins;-Interactome analysis of proteins that are simultaneously included in all three studied proteomes using STRING database.

### 2.2. Global Disorder Analysis of Proteomes of A-Bodies and nSBs

The proteomes of A-bodies formed in response to heat shock or acidosis, and heat-induced nSBs were subjected to the global disorder analysis, and a set of commonly utilized disorder predictors was used to evaluate the intrinsic disorder predispositions of these organelles. The results of these analyses are shown in Figure 2. 

Overall, this analysis reveals high levels of disorder in the proteomes of these MLOs. In fact, Figure 2A shows the results of the classification of the disorder status of these proteins based on the outputs of the per-residue disorder predictor PONDR^®^ VSL2. This classification is based on the practice that is accepted in the field to group proteins based on their PPDR values (where the percentage of the predicted disordered residues (PPDR) reflects the content of residues with disorder scores exceeding the 0.5 threshold) [32]. In this classification, proteins with PPDR of <10% are considered as ordered or mostly ordered; proteins with 10% ≤ PPDR < 30% are considered as moderately disordered; and proteins with PPDR of ≥30% are considered as highly disordered [32]. 

While conducting the analysis of the intrinsic disorder predisposition of query proteins, in addition to checking their PPDR levels, one should also look at the mean disorder scores (MDS) of the query proteins. This is because for a given protein, the MDS value does not always represent a direct reflection of its PPDR value. In fact, one can imagine a situation where a protein with a PPDR of 100% might have an MDS ranging from 0.5 to 1.0, whereas a protein with a PPDR of 0% might have an MDS of < 0.5. Taking MDS into account, proteins can be classified as highly ordered (MDS < 0.15), moderately disordered or flexible (MDS between 0.15 and 0.5) and highly disordered (MDS ≥ 0.5). Based on these classification criteria, none of the proteins analyzed in this study were predicted as ordered by either MDS or PPDR.

In the proteomes of the acidosis- and heat-induced A-bodies, there were only 1.1% and 1.83% of proteins predicted as mostly ordered based on their MDS values, respectively, whereas nSBs did not contain such proteins. Therefore, the vast majority of proteins in A-bodies and all proteins in nSBs can be classified as either moderately or highly disordered. In fact, Figure 2A shows that the proteomes of the heat-induced A-bodies, acidosis-induced A-bodies, and NsBs contained 21.1%, 26.22%, and 7.5% moderately disordered/flexible proteins (or proteins containing noticeable intrinsically disordered regions (IDPRs) based on both their MDS and PPDR values (these are proteins located within the dark pink segment)), respectively. An additional 17.8%, 20.73%, and 13.5% of these proteomes were expected to be moderately disordered based on their MDS values (light pink segment), whereas 60.0%, 51.22%, and 78.9% of proteins in the proteomes of acidosis-induced A-bodies, heat-induced A-bodies, and nSBs, respectively, were expected to be highly disordered, being located within the red segment (see Figure 2A).

At the next stage, we utilized a ΔCH-ΔCDF plot (a tool that combines the outputs of two binary predictors, i.e., tools classifying proteins as mostly ordered or mostly disordered), charge-hydropathy (CH) plot, and cumulative distribution function (CDF) plot (see Section 4 for more detail), to gain further information on the global disorder status of query proteins. Based on their position within the ΔCH-ΔCDF phase space, proteins can be classified as mostly ordered, molten globule-like or hybrid, or highly disordered. Figure 2B shows that 35.1%, 37.2%, and 22.6% of proteins in the proteomes of the acidosis-induced A-bodies, heat-induced A-bodies, and nSBs, respectively, are located within the bottom right corner that contains proteins predicted as ordered by both predictors. On the other hand, 30.3%, 25.6%, and 19.5% of these proteins are positioned within the bottom left corner, where one can find either compact but disordered molten globular proteins, or hybrid proteins containing high levels of ordered and disordered residues (i.e., proteins predicted to be ordered/compact by the CH-plot and disordered by the CDF analysis). 

In the top left corner, there are 32.4%, 32.9%, and 53.4% of proteins that are expected to be highly disordered and behave as native coils or native pre-molten globules in their unbound states, being predicted as disordered by both predictors. Finally, 2.2%, 4.3%, and 4.5% of proteins in the proteomes of the acidosis-induced A-bodies, heat-induced A-bodies, and nSBs, respectively, are predicted as disordered by CH-plot and ordered by CDF analysis (these are located within the top right corner). 

Taken together, the data shown in Figure 2 indicate that more than two-thirds of proteins from the three MLOs analyzed in this study are expected to contain high disorder levels. Importantly, based on the numbers shown in Figure 2, these MLOs are expected to be significantly more disordered than the human proteome [33]. In fact, a similar PPDR-MDS analysis previously conducted for 20,317 manually curated human proteins revealed higher levels of mostly ordered proteins (5.1%) and noticeably lower levels of highly disordered proteins (39.8%). Similarly, based on the results of CH-CDF analysis, the human proteome was shown to contain a significantly higher number of ordered proteins (59.1%) and significantly smaller number of disordered proteins (12.3%), as compared to the 35.1%, 37.2%, and 22.6% of ordered and 32.4%, 32.9%, and 53.4% of disordered proteins in the proteomes of the acidosis-induced A-bodies, heat-induced A-bodies, and nSBs, respectively. 

### 2.3. Global Analysis of LLPS Predisposition of Proteomes of A-Bodies and nSBs

Next, we analyzed the global LLPS predisposition of proteins in the three MLOs of interest. To this end, we evaluated their predispositions to undergo spontaneous LLPS using the FuzDrop platform (https://fuzdrop.bio.unipd.it/predictor, accessed on 27 April 2023) [34,35,36] and compared them to the global disorder status of these proteins. Figure 3 represents the results of this analysis and shows that very significant parts of the analyzed proteomes are expected to be present by the disordered proteins capable of driving LLPS. In fact, 54.5%, 40.2%, and 75.2% of proteins in the acidosis-induced A-bodies, heat-induced A-bodies, and nSBs, respectively, are expected to have the probability of spontaneous liquid–liquid phase separation, p_LLPS_, exceeding the threshold of 0.6. Most of these proteins capable of spontaneous LLPS (85.0%, 81.8%, and 81.0%, respectively) were predicted to have PPDR values exceeding 50%, with almost all of these LLPS-prone proteins possessing PPDR values of at least 30%. 

It should be noted that all three MLOs analyzed here contained noticeably more LLPS-promoting proteins than the human proteome, where 37.2% of proteins were previously reported as proteins which can spontaneously phase separate [34]. Therefore, proteomes of the acidosis-induced A-bodies, heat-induced A-bodies, and nSBs contained 1.46-, 1.08-, and 2.02-times more “droplet-driving” proteins than the human proteome, respectively. Figure 3 also shows that although none of the mostly ordered (PPDR < 10%) or moderately disordered (10% ≤ PPDR < 30%) proteins in the three MLOs are predicted to promote LLPS, not all highly disordered proteins are expected to serve as droplet drivers. In fact, of the 144, 118, and 123 proteins with a PPDR of ≥30% in the acidosis-induced A-bodies, heat-induced A-bodies, and nSBs, 46 (31.9%), 53 (44.9%), and 23 (18.7%) were not predicted as “droplet-driving” proteins. Even among the 103, 67, and 99 proteins with a PPDR of ≥50%, 18 (17.5%), 14 (20.9%), and 8 (8.1%) were not identified as droplet promoters. These observations suggest that although most of the “droplet-driving” proteins are expected to be moderately or highly disordered, not all proteins with high levels of disorder can undergo spontaneous LLPS.

### 2.4. Heat-Shock-Induced A-Bodies

In their 2019 paper, Marijan et al. identified at least 164 proteins that make up A-bodies formed in the nuclei of MCF-7 cells in response to heat shock [18]. The performed analysis showed that regardless of the high proteome disorder (approximately 72% of proteins), only about 20% of the proteome (33 proteins) of such bodies can be attributed to proteins with an extremely high probability of spontaneous phase separation that exceeds 0.9 (Figure 4).

These are mainly RNA-binding proteins involved in RNA processing, nuclear matrix proteins, and components of the nuclear nucleoprotein complex (see Figure 5 and Figure 6).

Furthermore, approximately 50% of the proteins of the studied proteome are assigned by the FuzDrop predictor to the so-called droplet-client protein group, i.e., proteins that can be included into the already pre-formed condensates, or proteins that are potentially capable of induced or assisted (but not spontaneous) liquid–liquid phase separation (Figure 7).

In addition, 15 proteins from the proteome of the heat-shock-induced A-bodies are prone for aggregation. Basically, these are globular proteins with an ordered structure (Appendix A). According to the current model of A-bodies formation, the formation of these structures can be initiated by the complex coacervation of positively charged proteins and negatively charged rIGSRNAs. Indeed, about 65% of the proteins in the studied proteome of the heat-shock-induced A-bodies are positively charged (Figure 7). However, the total charge of the proteins that make up the proteome of the heat-shock-induced A-bodies is close to zero, and the most positively charged proteins are globular, hardly capable of LLPS.

### 2.5. Acidosis-Induced A-Bodies

Audas et al. revealed 185 proteins that are part of A-bodies formed under hypoxic conditions (1% O_2_, pH 6.0) in human MCF-7 cells [19]. It is known that the proteome of acidosis-induced A-bodies and heat-induced A-bodies overlaps by only 20% [18]. 

We have shown that the proportion of IDP and IDPRs in the proteome of acidosis-induced A-bodies is about 78%, which is 6% higher than the content of such proteins in heat-induced A-bodies (Figure 8A). At the same time, 66 proteins (35.7%) that make up the studied proteome are highly likely to be predisposed to spontaneous LLPS (Figure 8B), which is 16% higher than the content of such proteins in heat-induced A-bodies. As with heat-shock-induced A-bodies, most of the LLPS-prone proteins that make up acidosis-induced A-bodies are RNA-binding proteins involved in RNA processing and splicing (Figure 9 and Figure 10).

According to the FuzDrop analysis, the number of driver proteins that make up the proteome of the acidosis-induced A-bodies is more than 1.5 times greater than the number of client proteins capable of only induced interactions with LLPS partners (Figure 11). At the same time, approximately 30% of acidosis-induced A-body proteomes represent proteins that are predisposed to inducible LLPS. This is significantly smaller than the content of such proteins in the heat-shock-induced A-body proteome.

The proportion of the proteins predisposed for aggregation in the acidosis-induced A-body’s proteome (7.5%) is even lower than that found in the heat-induced A-bodies (Appendix A). At the same time, it seems that the proportion of positively charged proteins in the A-body proteome does not depend on the type of stress effect on the cell.

### 2.6. Heat-Induced Nuclear Stress Bodies (nSBs)

Ninomiya et al. identified 141 protein species that are part of nSBs (i.e., protein species associated with HSATIII as detected by the ChIRP-MS method), which are formed under heat shock conditions in human HeLa cells [28]. Since, in addition to the individual proteins, the reported data contained information on the protein complexes or proteins of the same family, some redundancy was present in the original dataset. This redundancy was reduced by keeping just one entry with a given protein ID. As a result of the removal of duplicated entries, a set of 131 unique entries was retrieved. Two nSB marker proteins, HSF1 and HSF2, were also included in this set, resulting in a total compendium of 133 proteins. According to our analysis, the nSB proteome is formed by highly disordered and LLPS-related proteins (Figure 12A). Over 90% of the nSB proteome contains highly disordered proteins, and 66% of the nSB proteome is formed by proteins with a high probability of prone LLPS.

These are mainly RNA-binding proteins involved in mRNA processing, RNA splicing, and RNA export from the nucleus (Figure 13 and Figure 14).

According to the FuzDrop predictor, the majority of proteins prone to LLPS are represented by “driver” proteins: proteins capable of forming the framework of membrane-less organelles and initiating the formation of biomolecular condensates (Figure 15). More than 70% of the proteins that make up nSBs are positively charged, which indirectly correlates with the RNA-binding properties of the nSB proteome. Less than 4% of the proteins that make up the nSB proteome are prone to aggregation (Appendix A).

## 3. Discussion

A comparative analysis of the proteomes of A-bodies formed in response to heat shock and acidosis, and nuclear stress bodies, showed that the morphology and mechanical properties of A-bodies and nSBs correlates with the protein composition of these organelles. A-bodies are MLOs that transform into fibrillar structures during their biogenesis, contain significantly fewer proteins prone to LLPS, and have less highly disordered proteins compared to nSBs, which normally have liquid-droplet characteristics (Table 1).

The significant content of LLPS “client” proteins in the A-body’s proteome is consistent with the existing model for the formation of these organelles, according to which the formation of these organelles is initiated by the complex coacervation of rIGSRNAs with proteins. Apparently, rIGSRNA molecules play the role of LLPS “drivers” in the formation of A-bodies. The characteristics of the protein composition of nSBs suggest that the LLPS of the scaffold proteins of these organelles may be the initial stage in the formation of these bodies. It is known that HSF1, which initiates HSATIII transcription, is capable of reversible LLPS under heat shock conditions [37].

At the same time, according to our analysis, acidosis-induced A-bodies have a larger number of proteins that promote “liquid-droplet” properties compared to the heat-shock-induced A-bodies and occupy an intermediate position between the nSBs and heat-shock-induced A-bodies in almost all analyzed characteristics. It should be noted that even the number of proteins in common with nSBs is higher in acidosis-induced A-bodies (Figure 16). We also identified 10 proteins that are simultaneously included in all three studied proteomes.

These are mainly RNA-binding proteins with a high degree of disorder and a tendency to LLPS, acting as phase separation drivers, not prone to amyloid formation (Table 2).

These proteins are components of heterogeneous ribonucleoprotein complexes that control pre-mRNA processing (HNRNPH1, HNRNPA0, HNRNPM, and HNRNPK), and participate in splicing (CGI-74, RBM39, and DDX39), transcription, and ribosome biogenesis (DDX 39 and nucleolin). HspA8 and PPIA are chaperones required for correct protein folding.

Although providing in-depth characterization of these proteins is outside the scope of this study, some illustrative details of their structure, functional disorder, and LLPS predisposition are summarized in Figure 17, Figure 18, and Figure 19, respectively.

Figure 17 represents 3D models generated for these 10 shared proteins by AlphaFold [38,39]. In line with the intrinsic disorder predisposition analysis (see Table 2), seven of these proteins are predicted to have very high levels of disorder as evidenced by the presence of yellow and orange “noodles” and cyan regions with low per-residue confidence scores. 

**Figure 17 ijms-24-11007-f017:**
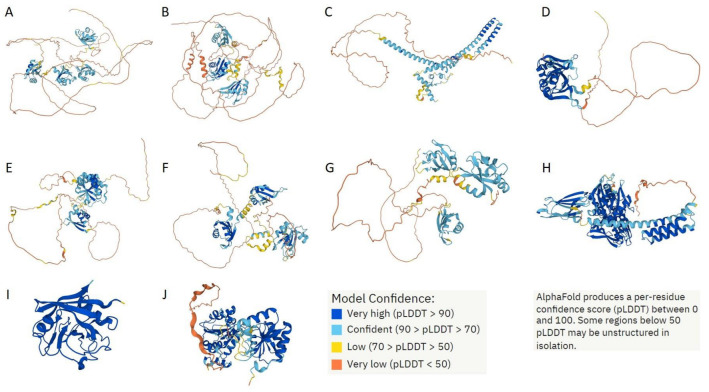
AlphaFold-generated 3D models of 10 proteins shared by the acidosis-induced A-bodies, heat-induced A-bodies, and nSBs. (**A**) NCL (P19338); (**B**) HNRNPM (P52272); (**C**) CGI-74 (Q9Y383); (**D**) HNRNPA0 (Q13151); (**E**) HNRNPK (P61978); (**F**) RBM39 (Q14498); (**G**) HNRNPH1 (P31943); (**H**) HSPA8 (P11142); (**I**) PPIA (P62937); (**J**) DDX39A (O00148). Proteins are ordered based on their intrinsic disorder propensity as evaluated by PONDR^®^ VSL2.

Figure 18 provides further support to the highly disordered status of most of these shared proteins and shows that in addition to high levels of disorder, they contain multiple disorder-based protein–protein interaction sites known as molecular recognition features (MoRFs), i.e., disordered regions capable of at least partial folding as a result of interaction with specific binding partners. 

**Figure 18 ijms-24-11007-f018:**
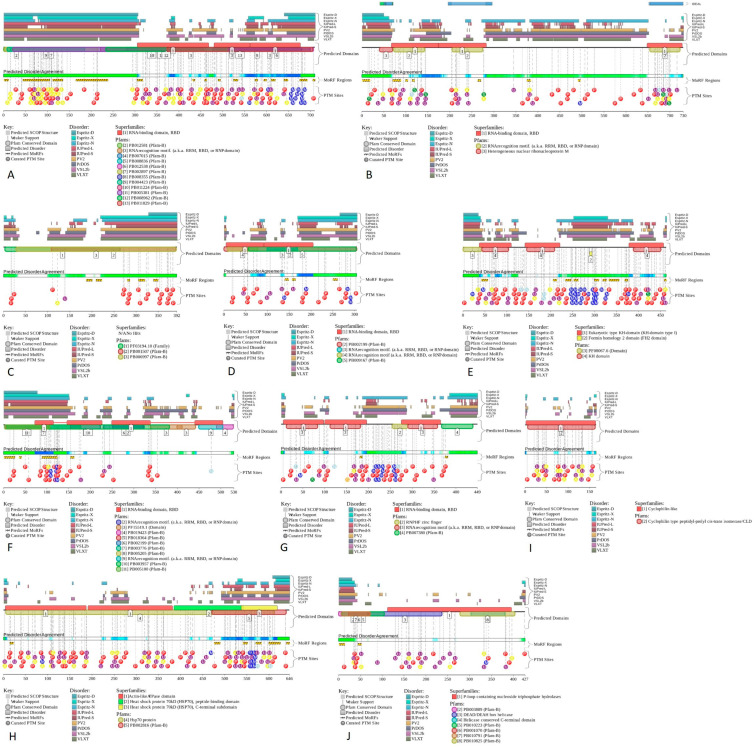
Functional disorder profiles of (**A**) NCL (P19338); (**B**) HNRNPM (P52272); (**C**) CGI-74 (Q9Y383); (**D**) HNRNPA0 (Q13151); (**E**) HNRNPK (P61978); (**F**) RBM39 (Q14498); (**G**) HNRNPH1 (P31943); (**H**) HSPA8 (P11142); (**I**) PPIA (P62937); and (**J**) DDX39A (O00148) generated by the D^2^P^2^ platform. Here, the IDPR localization predicted by IUPred, PONDR^®^ VLXT, PONDR^®^ VSL2, PrDOS, PV2, and ESpritz are shown by nine differently colored bars on the top of the plot, whereas the blue–green–white bar in the middle of the plots shows the agreement between the outputs of these disorder predictors, with disordered regions by consensus being shown by blue and green. The two lines with colored and numbered bars above the disorder consensus bar show the positions of functional SCOP domains [40,41], predicted using the SUPERFAMILY predictor [42]. Positions of the predicted disorder-based binding sites (molecular recognition feature (MoRFs)) identified by the ANCHOR algorithm are shown by yellow zigzagged bars [43]. Locations of the sites of different posttranslational modifications (PTMs) identified by the PhosphoSitePlus platform [44] are shown at the bottom of the plots by the differently colored circles. Larger-size functional profiles of these proteins can be found in the Appendix A.

Furthermore, all these proteins are heavily decorated by various posttranslational modifications (PTMs). Therefore, the 10 proteins shared by the acidosis-induced A-bodies, heat-induced A-bodies, and nSBs are typical IDPs or hybrid proteins containing functionally important IDPRs. It is likely that these disorder-based features define the multifunctionality of these proteins.

Figure 19 shows the distribution of the residue-based droplet promoting probabilities within the sequences of these 10 proteins. According to this analysis, nine of these proteins are capable either of spontaneous LLPS or can act as droplet clients. The only exception is PPIA, which is not predicted to contain droplet-promoting regions and shows a low p_LLPS_ value (see Table 2). 

**Figure 19 ijms-24-11007-f019:**
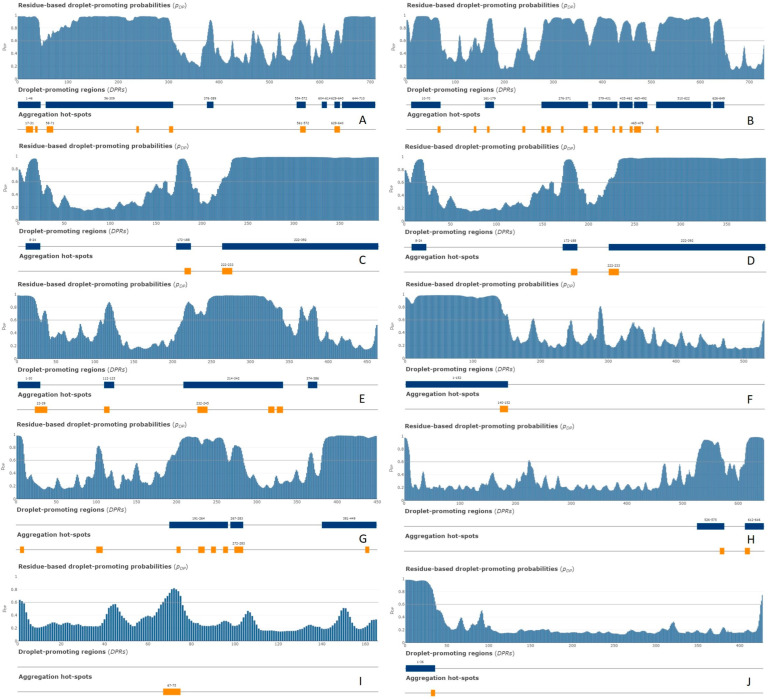
FuzDrop profiles of (**A**) NCL (P19338); (**B**) HNRNPM (P52272); (**C**) CGI-74 (Q9Y383); (**D**) HNRNPA0 (Q13151); (**E**) HNRNPK (P61978); (**F**) RBM39 (Q14498); (**G**) HNRNPH1 (P31943); (**H**) HSPA8 (P11142); (**I**) PPIA (P62937); and (**J**) DDX39A (O00148).

It should be noted that, according to the STRING database, these 10 proteins shared by the three MLOs analyzed in this study (i.e., acidosis-induced A-bodies, heat-induced A-bodies, and nSBs) are expected to have both functional and physical associations, creating a rather dense network with an average local clustering coefficient of 0.772 and an average node degree of 5.6, a network, where 10 query proteins are connected by 28 interactions/associations (Figure 20).

We also used the STRING platform to evaluate the global interactability of the resulting internal network, i.e., to create an external network that includes proteins associated with these 10 proteins. The corresponding network was generated using a custom confidence of 0.790 for the minimum required interaction, which was selected to include the maximum number of interactors (note that the number of interactors in STRING is limited to 500). The resulting interactome includes 494 nodes connected by 6907 edges. This interactome is characterized by an average node degree of 28 and shows an average local clustering coefficient of 0.653. The expected number of interactions for the set of proteins of this size is 2531, indicating that this PPI network centered on 10 proteins shared by acidosis-induced A-bodies, heat-induced A-bodies, and nSBs has significantly more interactions than expected (PPI enrichment *p*-value is <10^−16^). 

Among the most significant biological processes ascribed to this interactome are gene expression (GO:0010467; *p*-value = 4.63 × 10^−116^), RNA processing (GO:0006396; *p*-value = 8.30 × 10^−115^); the RNA metabolic process (GO:0016070; *p*-value = 5.13 × 10^−86^); mRNA processing (GO:0006397; *p*-value = 1.35 × 10^−85^); and RNA splicing (GO:0008380; *p*-value = 1.58 × 10^−84^). The five most significant molecular functions of this interactome are RNA binding (GO:0003723; *p*-value = 5.32 × 10^−132^); nucleic acid binding (GO:0003676; *p*-value = 9.42 × 10^−78^); heterocyclic compound binding (GO:1901363; *p*-value = 2.14 × 10^−66^); organic cyclic compound binding (GO:0097159; *p*-value = 1.52 × 10^−65^); and mRNA binding (GO:0003729; *p*-value = 8.74 × 10^−56^), whereas its five most significant cellular components are protein-containing complexes (GO:0032991; *p*-value = 3.23 × 10^−97^); ribonucleoprotein complexes (GO:1990904; *p*-value = 1.90 × 10^−89^); nucleoplasm (GO:0005654; *p*-value = 1.54 × 10^−87^); nuclei (GO:0005634; *p*-value = 4.86 × 10^−83^); and nuclear lumen (GO:0031981; *p*-value = 2.13 × 10^−81^). 

Furthermore, this interactome can be divided into three well-separated clusters (Figure 21). A total of 227 proteins included into red cluster are linked by 2207 edges. This cluster has an average node degree of 19.4 and is characterized by an average clustering coefficient of 0.744. Its proteins are involved in the following biological processes: the regulation of cellular response to heat (GO:1900034; *p*-value = 1.83 × 10^−68^); protein folding (GO:0006457; *p*-value = 3.33 × 10^−55^); the transport of viruses (GO:0046794; *p*-value = 9.02 × 10^−54^); the intracellular transport of viruses (GO:0075733; *p*-value = 2.30 × 10^−53^); and the establishment of localization in the cell (GO:0051649; *p*-value = 6.47 × 10^−53^). Approximately 27% of proteins of the studied proteomes are included in this cluster.

Their most significant molecular functions are heat shock protein binding (GO:0031072; *p*-value = 1.55 × 10^−37^); unfolded protein binding (GO:0051082; *p*-value = 1.28 × 10^−33^); chaperone binding (GO:0051087; *p*-value = 9.58 × 10^−33^); protein binding (GO:0005515; *p*-value = 1.22 × 10^−29^); and the structural constituent of the nuclear pore (GO:0017056; *p*-value = 3.19 × 10^−24^). The most significant cellular components of proteins from this cluster are cytosol (GO:0005829; *p*-value = 4.51 × 10^−41^); the host cell (GO:0043657; *p*-value = 4.21 × 10^−39^); protein-containing complexes (GO:0032991; *p*-value = 2.12 × 10^−37^); the endomembrane system (GO:0012505; *p*-value = 4.72 × 10^−32^); and the nuclear pore (GO:0005643; *p*-value = 2.26 × 10^−30^).

The green cluster has 140 proteins connected by 1904 edges. This cluster is characterized by an average node degree of 27.2 and average local clustering coefficient of 0.704. The most significant biological processes here are mRNA processing (GO:0006397; *p*-value = 1.39 × 10^−135^); RNA splicing (GO:0008380; *p*-value = 1.80 × 10^−128^); RNA processing (GO:0006396; *p*-value = 5.56 × 10^−116^); RNA splicing, via transesterification reactions (GO:0000375; *p*-value = 5.52 × 10^−108^); and mRNA splicing, via spliceosome (GO:0000398; *p*-value = 1.68 × 10^−106^). The most significant molecular functions of proteins in the green cluster are RNA binding (GO:0003723; *p*-value = 2.04 × 10^−114^); nucleic acid binding (GO:0003676; *p*-value = 1.60 × 10^−80^); mRNA binding (GO:0003729; *p*-value = 1.63 × 10^−68^); Binding (GO:0005488; *p*-value = 3.86 × 10^−23^); and pre-mRNA binding (GO:0036002; *p*-value = 1.92 × 10^−21^). The most significant cellular components are nucleoplasm (GO:0005654; *p*-value = 2.29 × 10^−71^); spliceosomal complexes (GO:0005681; *p*-value = 2.04 × 10^−69^); ribonucleoprotein complexes (GO:1990904; *p*-value = 4.14 × 10^−66^); nuclear lumen (GO:0031981; *p*-value = 4.21 × 10^−64^); and nuclear speck (GO:0016607; *p*-value = 9.89 × 10^−57^). 

Finally, the 127 proteins assembled into the blue cluster are linked by 1494 edges. This cluster has an average node degree of 23.5 and is characterized by an average local clustering coefficient of 0.778. The most significant biological processes, in which these proteins take part are rRNA processing (GO:0006364; *p*-value = 1.77 × 10^−60^); ribosome biogenesis (GO:0042254; *p*-value = 9.60 × 10^−58^); ribonucleoprotein complex biogenesis (GO:0022613; *p*-value = 5.17 × 10^−56^); ncRNA processing (GO:0034470; *p*-value = 2.15 × 10^−55^); and RNA processing (GO:0006396; *p*-value = 2.03 × 10^−46^). Their most significant biological functions are RNA binding (GO:0003723; *p*-value = 2.75 × 10^−50^); nucleic acid binding (GO:0003676; *p*-value = 3.47 × 10^−33^); heterocyclic compound binding (GO:1901363; *p*-value = 9.10 × 10^−26^); organic cyclic compound binding (GO:0097159; *p*-value = 2.38 × 10^−25^); and snoRNA binding (GO:0030515; *p*-value = 6.51 × 10^−18^). These proteins are cellular components of small-subunit processomes (GO:0032040; *p*-value = 1.16 × 10^−49^); nucleolus (GO:0005730; *p*-value = 1.16 × 10^−49^); preribosomes (GO:0030684; *p*-value = 1.40 × 10^−49^); ribonucleoprotein complexes (GO:1990904; *p*-value = 2.15 × 10^−47^); and nuclear lumen (GO:0031981; *p*-value = 2.01 × 10^−37^).

Based on the results of our analysis, it follows that the machinery of A-bodies and nSBs aimed at mRNA processing, the preservation of cell protein material, transcription, and, possibly, nuclear translation, at least partially overlaps. At the same time, the regulation of these processes is clearly organelle-specific. Perhaps, this is due to the launch of specific stress response programs in specific stressful conditions.

Finally, we have to emphasize that one of the shortcomings of this study is based on the fact that in our analyses of proteins located in A-bodies and nSBs, equal weights were given to each individual protein present in the MLO. However, it is possible that levels of different proteins located within MLOs could be different, with some of them being more abundant than others. Since it is difficult to factor in the copy numbers for each protein species in an MLO, this represents a shortcoming that makes the conclusions regarding the overall properties of MLOs less certain.

## 4. Materials and Methods

### 4.1. Assembly of the Datasets for the Analysis

Corresponding datasets were assembled based on the previously published data on the proteome analysis of heat-shock-induced A-bodies [19], acidosis-induced A-bodies [18], and heat-shock-induced nuclear stress bodies [28].

### 4.2. Intrinsic Disorder Analysis

The disorder analysis of the studied proteins was carried out using the RIDAO online service [45] to predict disordered regions in proteins based on their amino acid sequences. This package combines the forecasts of several well-known disorder predictors, such as PONDR^®^ VLXT [46], PONDR^®^ VL3 [47], PONDR^®^ VLS2B [48], PONDR^®^ FIT [49], IUPred2 (Short), and IUPred2 (Long) [50,51]. In the case of a significant discrepancy between the results obtained by different predictors, the correctness of the protein structure disorder prediction was determined by the protein structure prediction using the AlphaFold2 algorithm [39]. 

The most consistent predictions of disorder for a large data set were generated by the PONDR^®^ VSL2B algorithm. As a measure of the disorder of the protein structure, we used the percentage of predicted disordered residues (PPDR_VSL2_), i.e., the percentage of residues in the protein amino acid sequence for which the predicted disorder score is higher than the threshold of 0.5. Such regions are defined by the algorithm as “probably disordered”. 

Next, we carried out a classification of proteins according to their degree of disorder. Proteins with PPDR_VSL2_ of < 10% were considered as ordered or mostly ordered; proteins with 10% ≤ PPDR_VSL2_ < 30% were considered as moderately disordered; and proteins with the PPDR_VSL2_ of ≥30% were considered as highly disordered [32]. Furthermore, proteins can be also classified based on their mean disorder scores (MDS) as highly ordered (MDS < 0.15), moderately disordered or flexible (MDS between 0.15 and 0.5), and highly disordered (MDS ≥ 0.5).

Next, the outputs of two binary predictors, the charge-hydropathy (CH) plot [52,53] and the cumulative distribution function (CDF) plot [53,54,55], were combined to conduct a CH-CDF analysis [55,56,57,58] that allows the classification of proteins based on their position within the CH-CDF phase space as ordered (proteins predicted to be ordered by both binary predictors), putative native “molten globules” or hybrid proteins (proteins determined to be ordered/compact by CH, but disordered by CDF), putative native coils and native pre-molten globules (proteins predicted to be disordered by both methods), and proteins predicted to be disordered by CH-plot, but ordered by CDF. 

Complementary disorder evaluations, together with important disorder-related functional information, were retrieved from the D^2^P^2^ database (http://d2p2.pro/, accessed on 27 April 2023) [59], which is a database of predicted disorder for a large library of proteins from completely sequenced genomes [59]. The D^2^P^2^ database uses outputs of IUPred [50], PONDR^®^ VLXT [46], PrDOS [60], PONDR^®^ VSL2B [47,61], PV2 [59], and ESpritz [62]. The visual console of D^2^P^2^ displays 9 colored bars representing the location of disordered regions as predicted by these different disorder predictors. In the middle of the D^2^P^2^ plots, the blue–green–white bar shows the predicted disorder agreement between nine disorder predictors (IUPred, PONDR^®^ VLXT, PONDR^®^ VSL2, PrDOS, PV2, and ESpritz), with the blue and green parts corresponding to the disordered regions by consensus. Above the disorder consensus bar are two lines with colored and numbered bars that show the positions of the predicted (mostly structured) SCOP domains [40,41] using the SUPERFAMILY predictor [42]. A yellow zigzagged bar shows the location of the predicted disorder-based binding sites (MoRF regions) identified by the ANCHOR algorithm [43], whereas differently colored circles at the bottom of the plot show the location of various PTMs assigned using the outputs of the PhosphoSitePlus platform [44], which is a comprehensive resource of the experimentally determined post-translational modifications.

The standard deviation calculated using the bootstrap was used as an error ID estimate. The subsample size was equal to the size of the set of proteins of the corresponding compartment; the number of generated subsamples was 10,000.

### 4.3. Analysis of the Interactability of Proteins

Information on the interactability of the 10 human proteins shared by the three MLOs was retrieved using the Search Tool for the Retrieval of Interacting Genes, STRING, http://string-db.org/ (accessed on 27 April 2023). STRING generates a network of protein–protein interactions based on predicted and experimentally validated information on the interaction partners [63]. In the corresponding network, the nodes correspond to proteins, while the edges show predicted or known functional associations. Seven types of evidence are used to build the corresponding network, which are indicated by the differently colored lines: a green line represents neighborhood evidence; a red line—the presence of fusion evidence; a purple line—experimental evidence; a blue line—co-occurrence evidence; a light blue line—database evidence; a yellow line—text mining evidence; and a black line—co-expression evidence [63]. 

In this study, STRING was utilized in two different modes: to generate the internal network protein–protein interactions (PPIs) between the 10 query proteins and to produce a global PPI network centered on these 10 proteins. The resulting PPI networks were further analyzed using STRING-embedded routines in order to retrieve network-related statistics, such as the number of nodes (proteins), the number of edges (interactions), average node degree (average number of interactions per protein), average local clustering coefficient (which defines how close the neighbors are to being a complete clique. If a local clustering coefficient is equal to 1, then every neighbor connected to a given node *N_i_* is also connected to every other node within the neighborhood, and if it is equal to 0, then no node that is connected to a given node *N_i_* connects to any other node that is connected to *N_i_*), expected number of edges (which is the number of interactions among the proteins in a random set of proteins of similar size), and a PPI enrichment *p*-value (which is a reflection of the fact that query proteins in the analyzed PPI network have more interactions among themselves than what would be expected for a random set of proteins of similar size, drawn from the genome. It was pointed out that such an enrichment indicates that the proteins are at least partially biologically connected, as a group).

The MCL algorithm was used to cluster the proteins that are shown in the resulting network.

### 4.4. LLPS Prediction

The studied datasets were further analyzed for the propensity of their proteins to undergo liquid–liquid phase separation (LLPS) using the FuzDrop [34] and PSPredictor [64] predictors. The propensity of the analyzed protein to LLPS was determined based on a PSPredcitor score of > 0.5 and FuzDrop score of > 0.6. In the event of a discrepancy between the results of predicting the propensity to phase separation obtained using both predictors, the analyzed proteins were assigned to the “controversial LLPS” group.

The FuzDrop predictor also predicts the ability of a query protein to undergo LLPS spontaneously (droplet-driving proteins) or if it requires additional interactions with LLPS partners to form droplets (droplet-clients). Proteins with a p_LLPS_ of ≥0.60 likely drive liquid–liquid phase separation. Proteins with a propensity for liquid–liquid phase separation (p_LLPS_) below the threshold of 0.6, but containing droplet-promoting regions (DPRs), defined as consecutive residues with a pDP of ≥0.60, will likely serve as droplet-clients.

### 4.5. Aggregation Propensity Prediction

Analysis of the propensity of the studied proteins to aggregation and the formation of amyloid fibrils was performed using the AggreScan package [65], which makes it possible to determine the so-called aggregation hot spots, i.e., areas of the protein that promote its aggregation. After preliminary analysis, it turned out that each protein in the studied datasets contained at least one such region. In this regard, the results of the performed analysis were supplemented using Normalized a4v Sequence Sum for 100 residues (Na4vSS) values, which reflect the average protein aggregation propensities of the sequences (Na4vSS > 0) once corrected for their size.

In addition, the ability of the analyzed proteins to form amyloid fibrils was verified by searching the CPAD database, which is composed of experimentally confirmed proteins prone to aggregation [66].

### 4.6. Calculation of Protein Charge and Hydrophobicity

The charge of the studied proteins was determined at pH = 7, according to [52], as the average protein charge over the sequence, taking into account the charge of amino acids: R = 1, K = 1, H = 0.5, D = −1, E = −1. The hydrophobicity of the studied proteins was calculated according to the normalized Kite and Doolittle scale.

### 4.7. Definition of Biological Processes and Molecular Functions of Proteins

It was performed using the Enrichr resource (https://maayanlab.cloud/Enrichr/, accessed on 20 April 2023), and 10 terms with the lowest *p*-values were selected. The *p*-values were computed from the Fisher exact test, which is used to determine whether or not there are significant nonrandom associations between two categorical variables.

## 5. Conclusions

The comparative analysis of the proteomes of heat-shock-induced and acidosis-induced A-bodies and nSBs may suggest that the propensity of MLO proteins to phase separate impacts the morphological properties of these compartments. Future experiments will have to address this hypothesis directly in vitro or in vivo. It was shown that in the series of heat-shock-induced A-bodies, acidosis-induced A-bodies, and heat-shock-induced nuclear stress bodies in the proteome of these organelles, the content of intrinsically disordered proteins and proteins that serve as LLPS drivers increases, whereas the content of the proteins prone to aggregation decreases. The performed analysis allows us to suggest that mRNA processing and the maintenance of the native structure of sequestered proteins in these organelles may be controlled by the same cellular machinery.

## Figures and Tables

**Figure 1 ijms-24-11007-f001:**
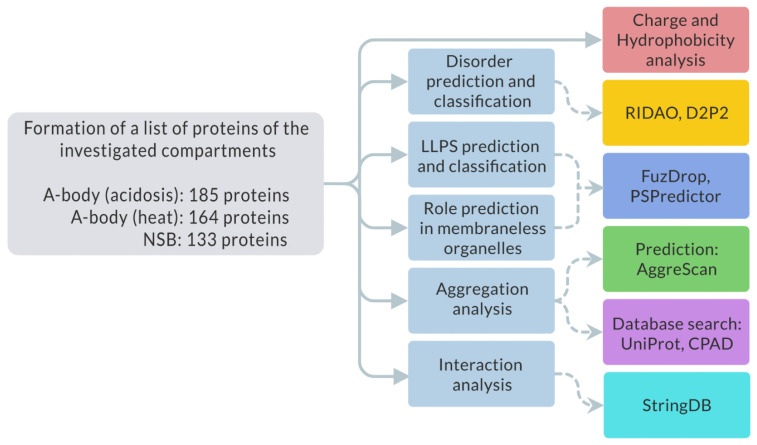
The scheme illustrating the analysis design.

**Figure 2 ijms-24-11007-f002:**
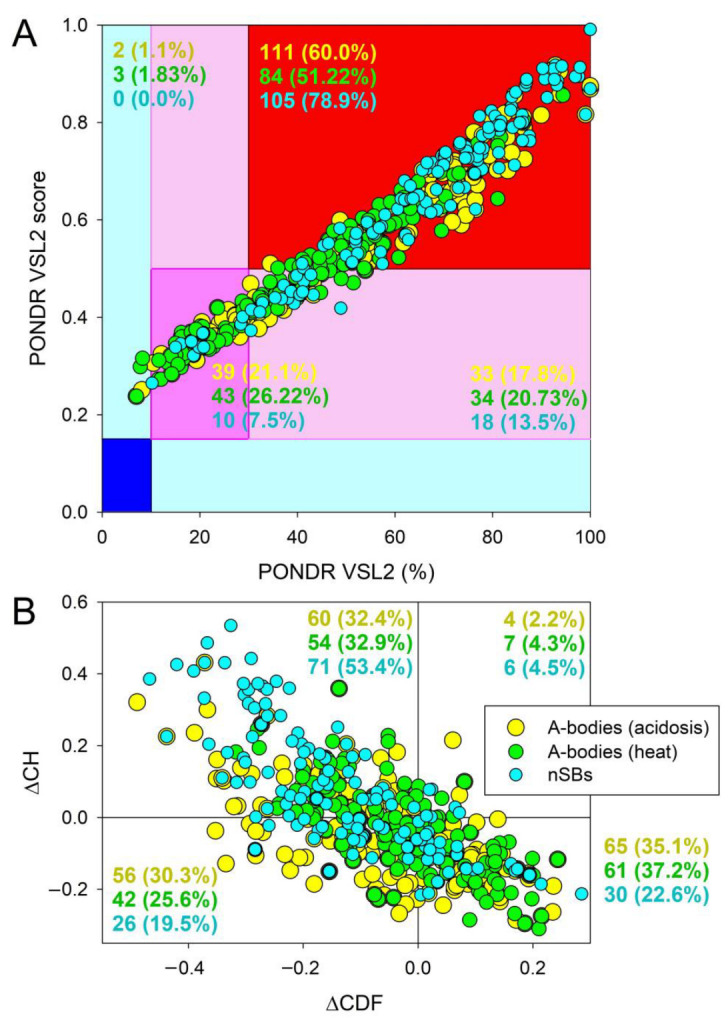
Evaluation of the global disorder predisposition of proteins in acidosis-induced A-bodies (yellow circles), heat-shock-induced A-bodies (green circles), and nSBs (cyan circles). (**A**) PONDR^®^ VSL2 output. PONDR^®^ VSL2 score is the mean disorder score (MDS) for a query protein. PONDR^®^ VSL2 (%) is a percentage of the predicted disordered residues (PPDR) in a query protein, i.e., the percentage of residues with disorder scores above 0.5. Color blocks indicate regions in which proteins are mostly ordered (blue and light blue), moderately disordered (pink and light pink), or mostly disordered (red), as per accepted classification (see the text). If the two parameters agree, the corresponding part of the background is dark (blue or pink), whereas light blue and light pink reflect areas in which only one of these criteria applies. (**B**) Charge-hydropathy and cumulative distribution function (CH-CDF) plot. The Y-coordinate is calculated as the distance of the corresponding protein from the boundary in the CH plot, whereas the X-coordinate is calculated as the average distance of the corresponding protein’s CDF curve from the CDF boundary. The quadrant in which the protein is located determines its classification. Q1, protein predicted to be ordered by CH-plot and CDF. Q2, protein predicted to be ordered by CH-plot and disordered by CDF-plot. Q3, protein predicted to be disordered by CH-plot and CDF. Q4, protein predicted to be disordered by CH-plot and ordered by CDF.

**Figure 3 ijms-24-11007-f003:**
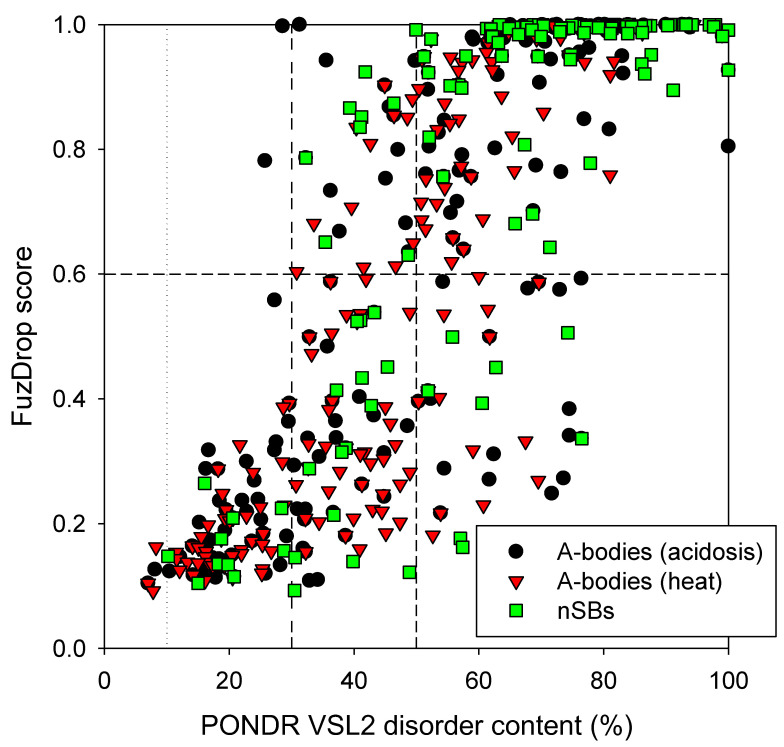
Comparison of spontaneous LLPS predisposition as evaluated by FuzDrop and propensity for intrinsic disorder as predicted by PONDR^®^ VSL2 of proteins in the proteomes of the acidosis-induced A-bodies, heat-induced A-bodies, and nSBs.

**Figure 4 ijms-24-11007-f004:**
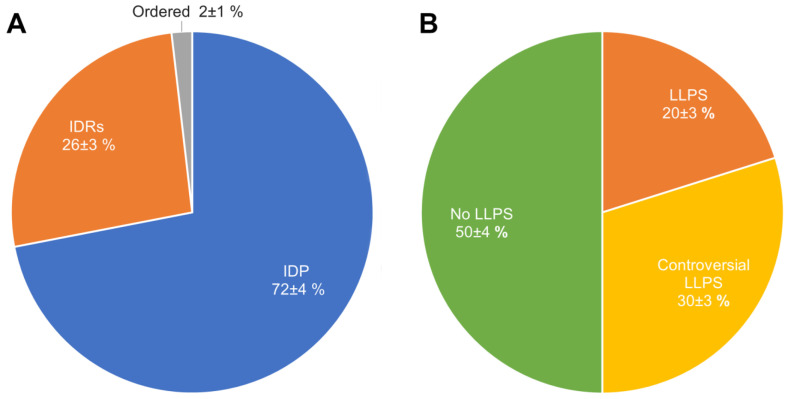
Pie charts representing the proportions of IDPs (Panel (**A**)) and LLPS-related proteins (Panel (**B**)) in the proteome of heat-shock-induced A-bodies according to the PONDR^®^ VSL2b and FuzDrop/PSP predictor analysis.

**Figure 5 ijms-24-11007-f005:**
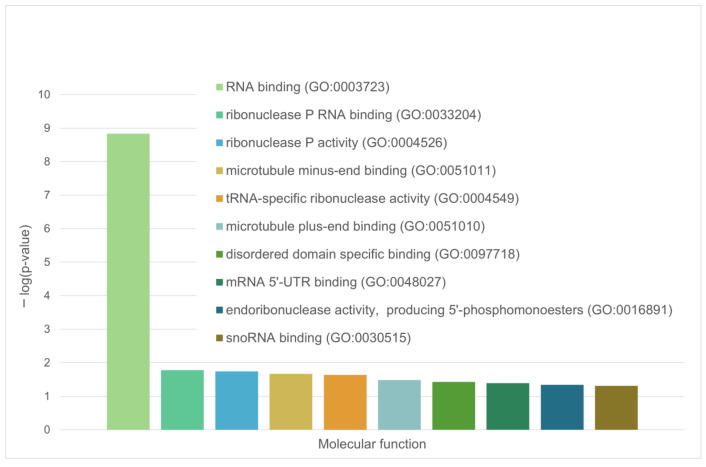
Diagram representing the molecular functions of LLPS-related proteins in the heat-shock-induced A-body’s proteome.

**Figure 6 ijms-24-11007-f006:**
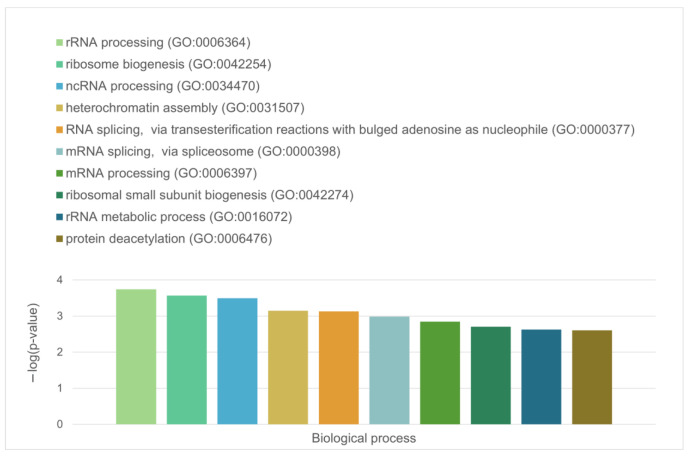
Diagram representing the biological processes that involve the LLPS-related proteins in the heat-shock-induced A-body’s proteome.

**Figure 7 ijms-24-11007-f007:**
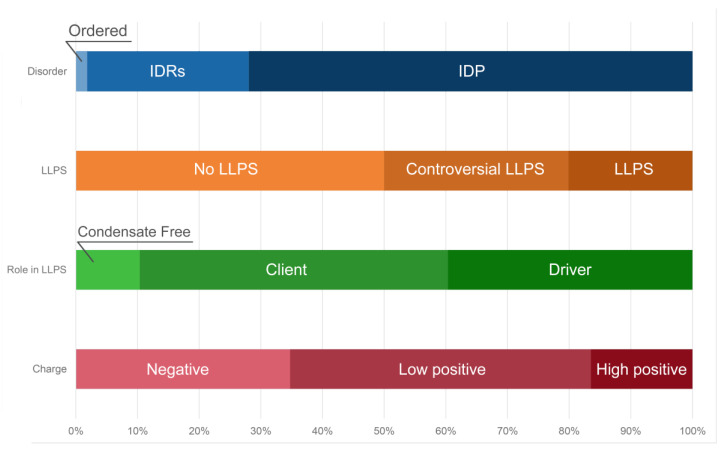
Distribution of the proteins from the heat-shock-induced A-body’s proteome based on protein disorder propensity, tendency to LLPS, and roles in LLPS according to FuzDrop, and protein charge.

**Figure 8 ijms-24-11007-f008:**
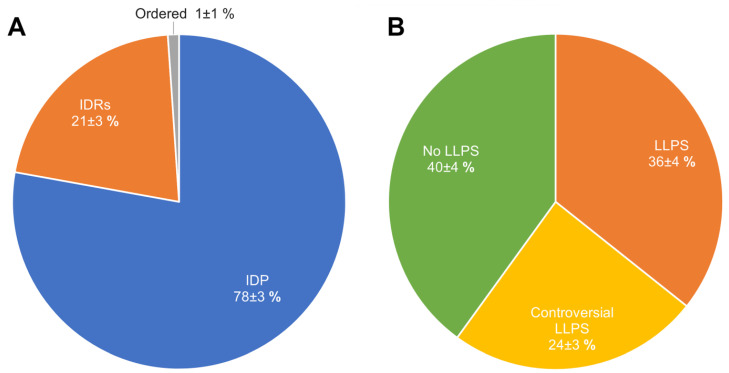
Pie charts representing proportions of IDPs (Panel (**A**)) and LLPS-related proteins (Panel (**B**)) in the proteome of the acidosis-induced A-bodies according to the PONDR^®^ VSL2b and FuzDrop/PSP predictor analysis, respectively.

**Figure 9 ijms-24-11007-f009:**
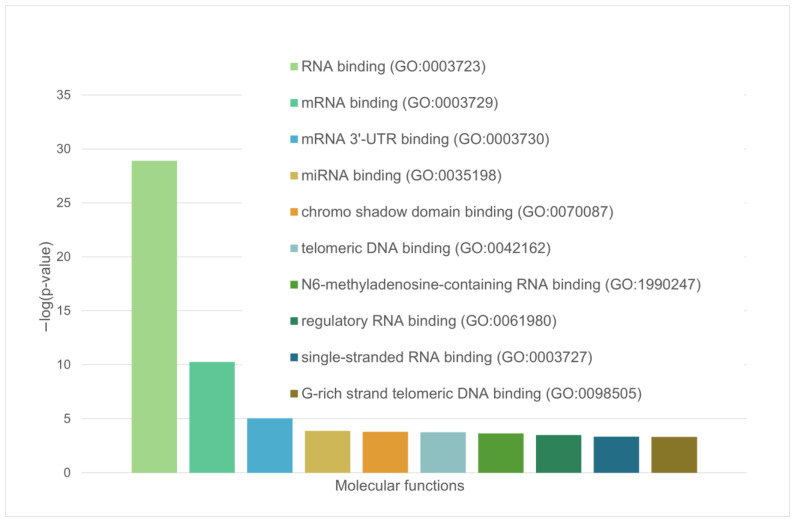
Diagram representing the molecular functions of LLPS-related proteins in acidosis-induced A-body’s proteome.

**Figure 10 ijms-24-11007-f010:**
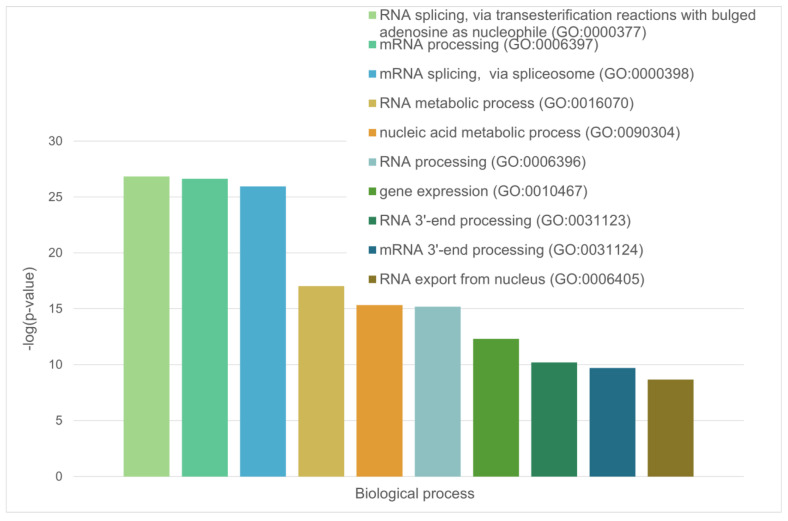
Diagram representing the biological processes that involve the LLPS-related proteins in acidosis-induced A-body’s proteome.

**Figure 11 ijms-24-11007-f011:**
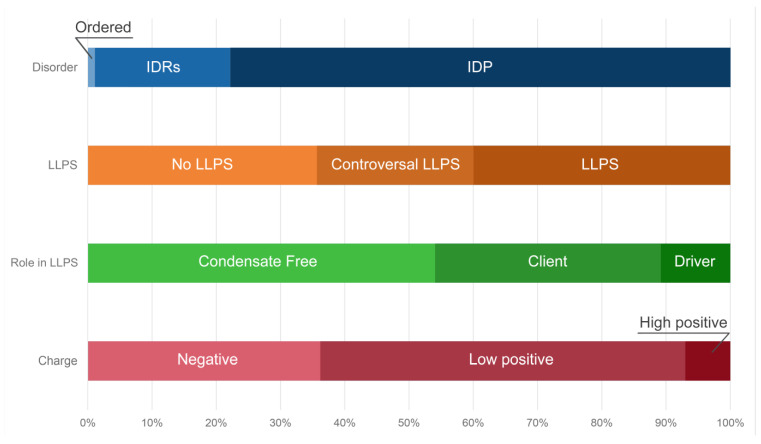
Distribution of acidosis-induced A-body’s proteome based on protein disorder, tendency to LLPS, roles in MLOs according to FuzDrop, and protein charge.

**Figure 12 ijms-24-11007-f012:**
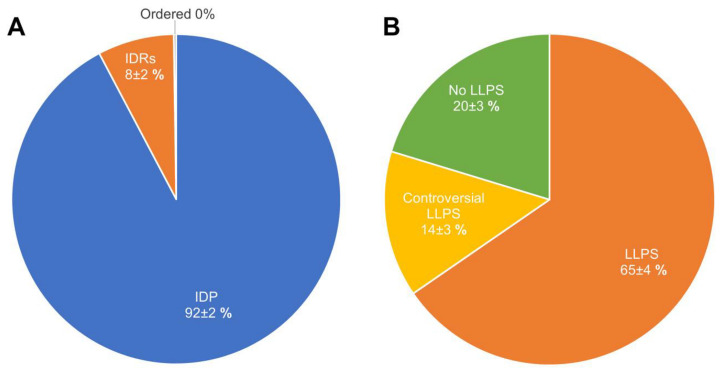
Pie charts representing the proportion of IDPs (Panel (**A**)) and LLPS-related proteins (Panel (**B**)) in the proteome of the nuclear stress bodies according to the PONDR^®^ VSL2b and FuzDrop/PSP predictor analysis, respectively.

**Figure 13 ijms-24-11007-f013:**
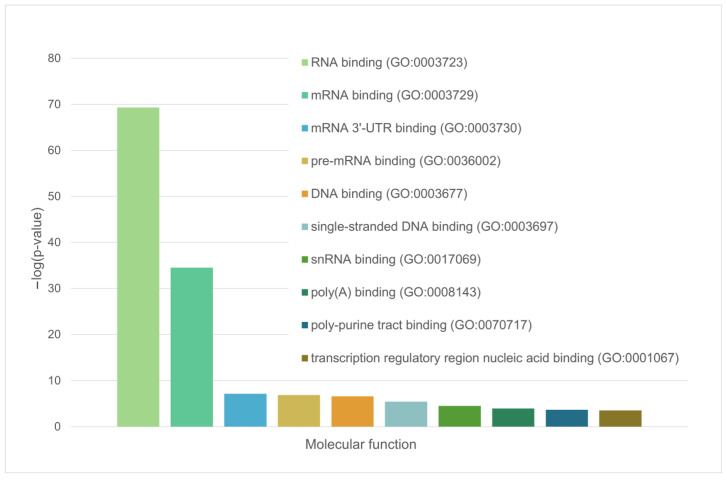
Diagram representing the molecular functions of LLPS-related proteins in the nuclear stress body’s proteome.

**Figure 14 ijms-24-11007-f014:**
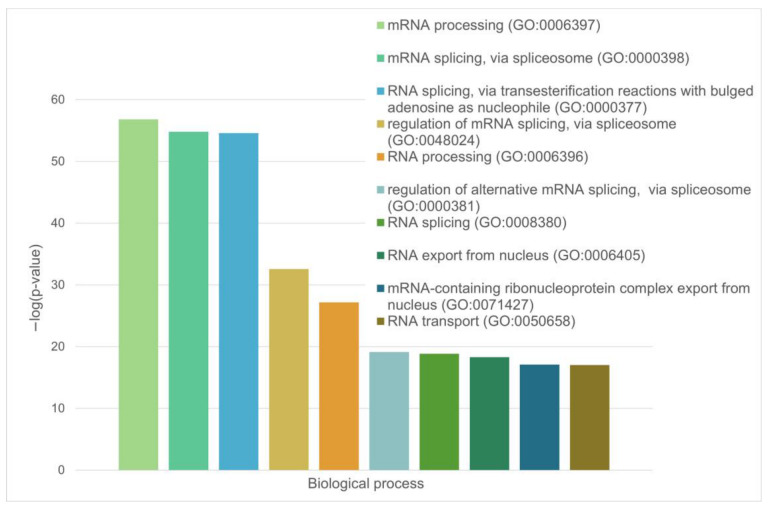
Diagram representing the biological processes that involve the LLPS-related proteins in the proteome of nuclear stress bodies.

**Figure 15 ijms-24-11007-f015:**
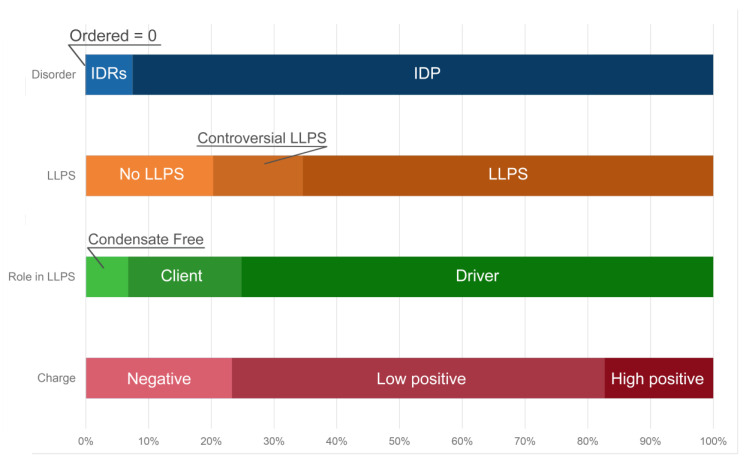
Distribution of protein intrinsic disorder, tendency to LLPS, roles in MLOs according to FuzDrop, and protein charge in the proteome of nSBs.

**Figure 16 ijms-24-11007-f016:**
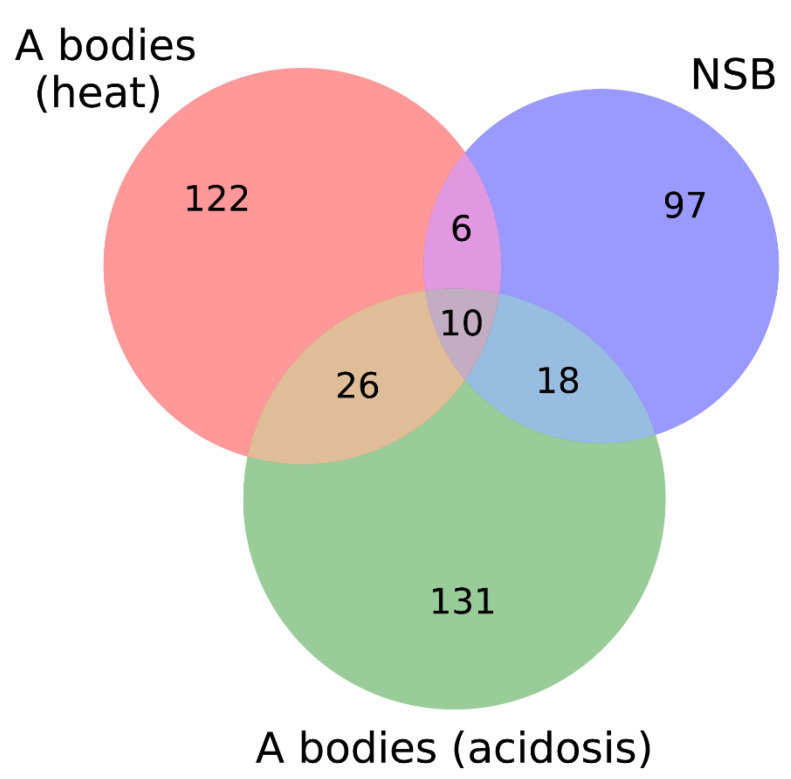
Venn diagram depicting the number of intersecting proteins between A-bodies and nSBs.

**Figure 20 ijms-24-11007-f020:**
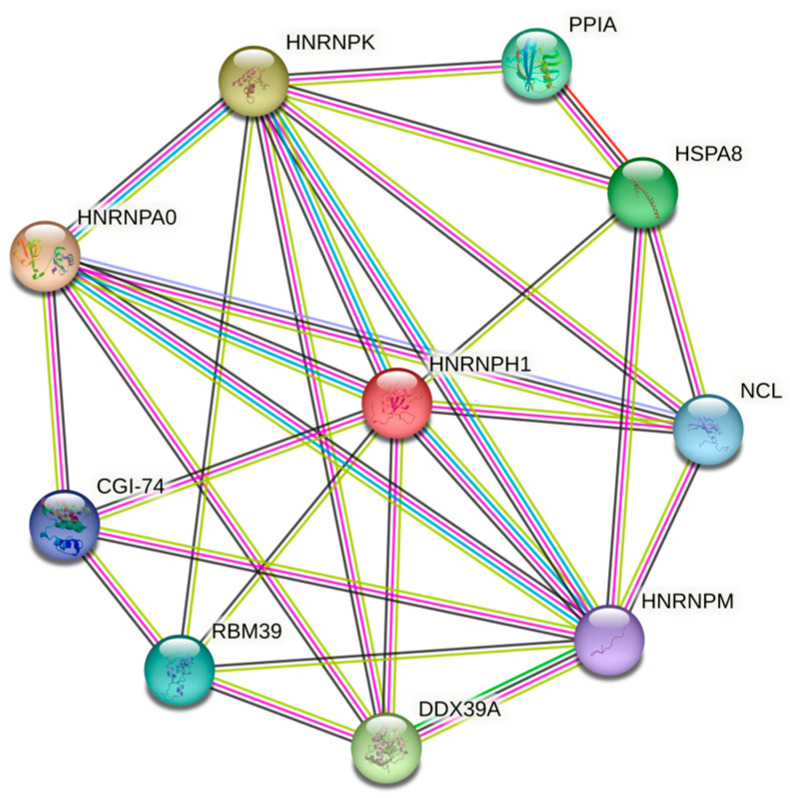
Network of intersecting proteins between A-bodies and nSBs according to STRING. The network was generated using a medium confidence of 0.400 for the minimum required interactions. Here, edges represent protein–protein associations that indicate that proteins jointly contribute to a shared function. However, the presence of such associations does not necessarily mean that these proteins are involved in physical interactions with each other. Edges are colored based on the evidence, where known interactions from curated databases and experimentally determined interactions are shown by cyan and pink lines, correspondingly. Predicted interactions based on the gene neighborhood, gene fusions, and gene co-occurrence are shown by green, red, and blue lines. Other types of evidence extracted based on text-mining, co-expression, and protein homology are shown by dark yellow, black, and violet lines, respectively.

**Figure 21 ijms-24-11007-f021:**
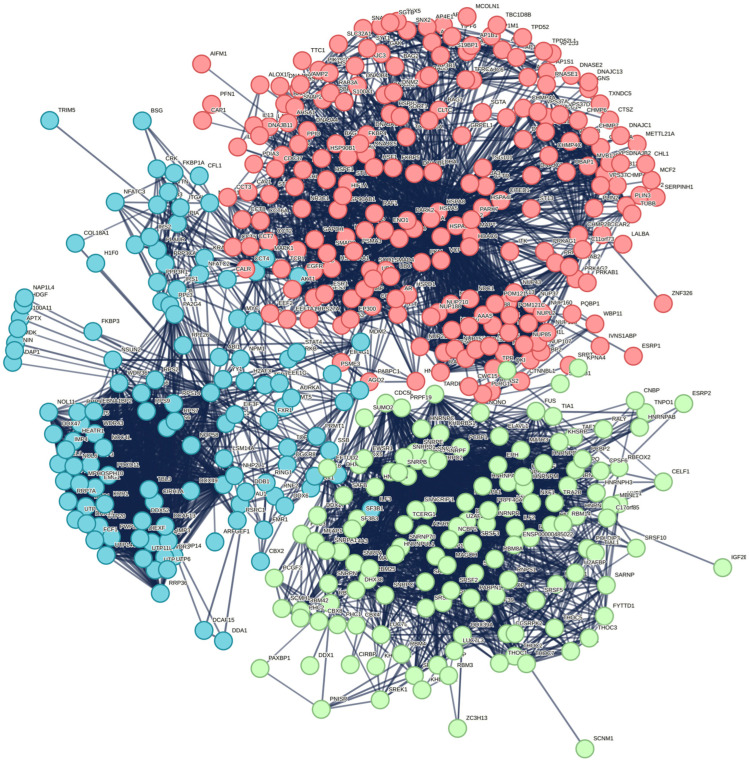
Interactome of intersecting proteins between A-bodies and nSBs according to STRING. Colors representing proteins clusters. There are 227, 140, and 127 proteins in the red, green, and blue clusters, respectively.

**Table 1 ijms-24-11007-t001:** Comparative characteristics of the proteomes of the heat-shock-induced and acidosis-induced A-bodies and nSBs.

	A-Bodies (Heat Shock)	A-Bodies (Acidosis)	nSBs (Heat Shock)
Proteome size	164	185	133
IDPs, %	72	78	92
LLPS-related proteins	33 (20.1%)	66 (35.7%)	89 (66.9%)
Drivers (FuzDrop)	65 (39.6%)	100 (54.1%)	102 (76.9%)
Clients (FuzDrop)	82 (50%)	65 (35.1%)	24 (18%)
Aggregation-related proteins	17 (10.3%)	14 (7.5%)	5 (3.8%)
Average proteome charge	0.02	0.01	0.03

**Table 2 ijms-24-11007-t002:** Comparative characteristics of the intersecting proteins between the A-bodies and nSBs. The green color represents characteristics according to which the analyzed protein has a high tendency for spontaneous LLPS. DPR are droplet-promoting regions, nHS means number of aggregation hot spots, and Na4vSS means Normalized a4v Sequence Sum for 100 residues used for aggregation prediction.

Gene Name (UniProt ID)	PER(VSL2b)	FuzDrop Score	PSPredictor Score	FuzDrop and PSP LLPS	DPRs	Role in MLOs	nHS	Na4vSS	Molecular Function
*NCL* (P19338)	86.2	0.9872	0.941	+	7	driver	17	−45.4	RNA binding
*HNRNPM* (P52272)	76.3	0.9997	0.8675	+	8	driver	18	−14	RNA binding
*CGI-74* (Q9Y383)	74.49	0.9512	0.0666	±	3	driver	2	−50.2	RNA binding
*HNRNPA0* (Q13151)	61.97	0.9818	0.9959	+	1	driver	5	−20.5	RNA binding
*HNRNPK* (P61978)	56.8	0.9031	0.2392	±	4	driver	9	−19.7	RNA binding
*RBM39* (Q14498)	51.89	0.4127	0.5286	±	1	client	17	−18.5	RNA binding
*HNRNPH1* (P31943)	51.22	0.9486	0.9727	+	3	driver	9	−16	RNA binding
*HSPA8* (P11142)	38.7	0.3211	0.5824	±	2	client	17	−13.8	RNA binding
*PPIA* (P62937)	20.61	0.2089	0.0057	−	0	no	7	−6.7	RNA binding
*DDX39A* (O00148)	18.27	0.1354	0.0079	−	1	client	16	−4	RNA binding

## Data Availability

The data present in the current study are available from the corresponding authors on reasonable request.

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
