# Peer review of "Nucleolar- and Nuclear-Stress-Induced Membrane-Less Organelles: A Proteome Analysis through the Prism of Liquid–Liquid Phase Separation"

_ijms, 2023, doi:10.3390/ijms241311007_

Round 1

Reviewer 1 Report

IDP and IDR play important roles in the formation of MLOs. This manuscript presents a comprehensive analysis of the three proteomes of MLOs, including the heat shock-induced and acidosis-induced A-bodies and nSBs. The results reveal that these proteomes contain a higher percentage of disordered regions and proteins than the average. Further analysis demonstrates that the machinery of these MLOs shares biological processes while exhibiting different regulation processes. The results are presented clearly and in detail.

Regarding the manuscript, I recommend that Figures 18 and 19 display only one or two proteins, with the others included in the supplement. Additionally, Figure 19 should have a more detailed legend.

There are several typos in the manuscript:

  1. Line 220: 'of proteins in the in the proteomes' should be corrected to 'of proteins in the proteomes.'
  2. Line 222: 'that I expected' should be corrected to 'that are expected.'
  3. Line 519: 'Thereare 227' should be corrected to 'There are 227.'
  4. Line 610: '10mhuman' should be corrected to '10 human.'

Author Response

Reviewer #1

Comments and Suggestions for Authors

The authors present a detailed bioinformatic study of the proteomes known for three membrane-less organelles, trying to connect the results with the morphological and functional properties of the organelles. The data presented by the authors are interesting and timely. The
manuscript is in general written clearly, with some typos and unclear
sentences. My main concerns are the lack of error bars in the reported percentage fractions, and the lack of details, plus some apparent inconsistencies, in the used methodologies.

Reply. We are grateful to Reviewer 1 for carefully reading our work and providing valuable comments. We hope that we were able to significantly improve the text quality.

Major comments

- authors should provide an estimate for the error of all the reported percentage fractions, for example by means of standard bootstrap sampling

Reply. We estimated error of reported protein fractions using means of standard bootstrap sampling

- variability in disorder prediction thresholds: it is really not clear why so many different thresholds are used for PONDR VSL2 disorder content; 10-30% in Fig. 1; and 30-50% in Fig. 2; and 50-85% to define the classes for pie-charts in Fig. 3 (and similar ones) or the bar plots in Fig. 6 (and similar ones); although in the text commenting Fig. 3 and similar ones the results related to the 80% threshold are reported. Furthermore, the two sentences at lines 283-286 "This analysis indicates that the structure of only 7 proteins out of the entire studied proteome of the heat-induced A-bodies is more than 80% disordered. This is consistent with the data that only about 20% of the proteome (33 proteins) of such bodies can be attributed to proteins with an extremely high probability of spontaneous phase separation that exceeds 0.9" are inconsequential in my opinion. Moreover it is not clear to which LLPS predictor the 0.9 threshold refers to.

Reply .We have led thresholds of protein disorder to definition the classes for the pie-chart along the full-text. Also, we eliminated the 0.9 threshold for LLPS prediction

- aggregation propensity: a first issue is that the results mentioned in the text (commenting Figure 6 and similar ones) are actually not present at all in the bar plots of Figure 6 (and similar ones), and that the CPAD database mentioned in Materials and Methods is actually never mentioned in the main text. A second, in my view more relevant issue, is that the Aggrescan predictor (similarly to what other "general" aggregation propensity predictors would do) is simply picking up the (few) proteins with an ordered structure within the large majority of mostly disordered proteins in MLO. The more prone to form native structure, the more prone to form amyloid structure is a general rule. I think it would be more relevant and much more interesting to check the results for aggregation propensity predictor in the context of prion-like domains, which should be specific for mostly disordered proteins. Finally, the exact definition of the Na4vSS score should be provided; in particular at line 659 it is referred to as a positive score, whereas in table 2 all negative scores are reported.

Reply. The prediction of aggregation propensity is actually a complex, challenging, and difficult process. The majority of existing predictors do not allow sufficient prediction accuracy. That is why we use combination of Aggrescan and CPAD database. CPAD database contains information on proteins for which formation amyloid fibrils by them has been experimentally confirmed. In Aggrescan server, the Na4vSS score is really determined as a positive score, we improved mistake in the manuscript text. We have added aggregation propensity information in supporting files. Prion-like domains, as a rule, signal that protein is firstly prone to spontaneous LLPS (and FuzDrop take this in into account)

- Figures 4, 5 and similar ones: I strongly suggest the authors to consider the Bonferroni test to check the bias for multiple hypothesis testing

Reply. For the diagrams with protein molecular functions and biological processes, we did not perform multiple comparisons between groups of proteins belonging to different fractions. The reported p-values were computed from the Fisher exact test.

- Table 2: it is not clear what some of the reported features are; DPRs (maybe Droplet Promoting Regions? it should be stated explicitly and how/with which method they are computed) and nHS, which is never mentioned in the manuscript; Na4vSS is mentioned in Materials and Methods (but see above comment) yet it should be stated in table caption that it refers to an aggregation propensity score

Reply. We improved Table 2

-  confidence for minimal required interactions in network generation: in Figure 19 caption it is stated to be 0.400, whereas at line 480 it is stated to be 0.790; however, I think the two networks (the one in fig. 19 and the one in fig. 20) should be generated for consistency in the same way. Moreover, from what the authors state at lines 480-481, it looks like the confidence needs to be higher to generate a network with a larger number of interactions; this seems counterintuitive if the confidence refers to the minimal threshold for required interactions in the network (as I get from fig. 19 caption). Finally, the authors should mention how they perform the clustering of nodes to obtain the clusters shown in Fig. 20.

Reply. The custom confidence for studied interactome of intersecting proteins between A-bodies and nSBs was chosen 0.790 because it was necessary to limit the number of interactors that STRING creates with medium confidence 0.400 (the number of interactors in STRING is limited to 500). We added information about clustering.

- Two sentences at the end of the discussion section (line 552: the machinery of MLOS at least partially overlaps) and in the conclusions section (lines 677-678: mRNA processing and maintenance of the native structure of sequestered proteins in these organelles is provided by the same cellular machinery) are inconsistent with each other. I do thing the former one (in the discussion section) is more representative of the actual results shown by the authors

Reply. Corrected

Minor comments

- line 40: IDRs -> IDPs (I guess)

Reply. IDPRs are hybrid proteins containing noticeable intrinsically disordered regions

- line 147: RBP is not defined

Reply. Corrected

- Figure 1B: quadrants mentioned in the caption are not labeled in the plot

Reply. Corrected

- line 275: to which paper does "their 2019 paper" refer to? 

Reply. Corrected

- line 292: it is not clear to which protein set "these" refers to. All LLPS proteins (as from Figure 3) or just the ones with score > 0.9 (but which one? see the first major comment)

Reply. Corrected

- caption of several figures: what are exactly LLPS-related proteins? I guess the LLPS class show in Fig. 3 and similar ones, but it should be stated explicitly.

Reply. Corrected

- line 323: I do not understand the sentence "We have shown that the proportion of IDP and IDPRs in the proteome of acidosis-induced A-bodies is about 56%, which is 10% higher than the content of such proteins in heat-induced A-bodies". 56% (49+7 from Fig. 7A) should compare to 41% (39+2 from Fig. 3A)

Reply. Corrected

- line 382: A-bodies have a ORDERED structure (not disordered as written by the authors), i.e. fibrillar, in comparison with nSBs

Reply. Corrected

- line 411: Hsc70 should be changed to HspA8, I guess

Reply. Corrected

In several places some words seem to be missing. I will give only a couple of examples (the likely missing word capitalized, when I can guess it) but there are several ones

- abstract, lines 28-29 "many of WHICH with high potential for spontaneous LLPS"

Reply. Corrected

- the region between an ordered and a disordered state characterized by largest a minimal impact on such systems can significantly change their properties and evolution over time" (in this case it is hard for me to find a way to make the sentence correct)

Reply. Corrected

- introduction, lines 156-17 "spontaneous and phase separation that IS  induced by interactions with partners" (although here I would probably change in "phase separation that is both spontaneous and induced by interactions with partners")

Reply. Corrected

- line 219,  a word is missing after "highly"

Reply. Corrected

- line 222, singular person should be changed to plural

Reply. Corrected

- line 621, "inter-network" -> internal network (I guess)

Reply. Corrected

- I am not sure in which sense the authors use the word "Curiously" (a couple of times). It sounds odd to me, I would change it maybe with noticeably (but I am not really sure what the authors mean).

Reply. Corrected

- the wording "presenting identified" is used in the caption of several figures; it makes no sense to me

Reply. Corrected

There are for sure other cases of sentences with English issues to be fixed

Reply. We significantly improved the text quality.

Reviewer 2 Report

The authors present a detailed bioinformatic study of the proteomes known for three membrane-less organelles, trying to connect the results with the morphological and functional properties of the organelles.

The data presented by the authors are interesting and timely. The
manuscript is in general written clearly, with some typos and unclear
sentences. My main concerns are the lack of error bars in the reported percentage fractions,  and the lack of details, plus some apparent inconsistencies, in the used methodologies.

Major comments

- authors should provide an estimate for the error of all the reported percentage fractions, for example by means of standard bootstrap sampling

- variability in disorder prediction thresholds: it is really not clear why so many different thresholds are used for PONDR VSL2 disorder content; 10-30% in Fig. 1; and 30-50% in Fig. 2; and 50-85% to define the classes for pie-charts in Fig. 3 (and similar ones) or the bar plots in Fig. 6 (and similar ones); although in the text commenting Fig. 3 and similar ones the results related to the 80% threshold are reported. Furthermore, the two sentences at lines 283-286 "This analysis indicates that the structure of only 7 proteins out of the entire studied proteome of the heat-induced A-bodies is more than 80% disordered. This is consistent with the data that only about 20% of the proteome (33 proteins) of such bodies can be attributed to proteins with an extremely high probability of spontaneous phase separation that exceeds 0.9" are inconsequential in my opinion. Moreover it is not clear to which LLPS predictor the 0.9 threshold refers to.

- aggregation propensity: a first issue is that the results mentioned in the text (commenting Figure 6 and similar ones) are actually not present at all in the bar plots of Figure 6 (and similar ones), and that the CPAD database mentioned in Materials and Methods is actually never mentioned in the main text. A second, in my view more relevant issue, is that the Aggrescan predictor (similarly to what other "general" aggregation propensity predictors would do) is simply picking up the (few) proteins with an ordered structure within the large majority of mostly disordered proteins in MLO. The more prone to form native structure, the more prone to form amyloid structure is a general rule. I think it would be more relevant and much more interesting to check the results for aggregation propensity predictor in the context of prion-like domains, which should be specific for mostly disordered proteins. Finally, the exact definition of the Na4vSS score should be provided; in particular at line 659 it is referred to as a positive score, whereas in table 2 all negative scores are reported.

- Figures 4, 5 and similar ones: I strongly suggest the authors to consider the Bonferroni test to check the bias for multiple hypothesis testing

- Table 2: it is not clear what some of the reported features are; DPRs (maybe Droplet Promoting Regions? it should be stated explicitly and how/with which method they are computed) and nHS, which is never mentioned in the manuscript; Na4vSS is mentioned in Materials and Methods (but see above comment) yet it should be stated in table caption that it refers to an aggregation propensity score

-  confidence for minimal required interactions in network generation: in Figure 19 caption it is stated to be 0.400, whereas at line 480 it is stated to be 0.790; however, I think the two networks (the one in fig. 19 and the one in fig. 20) should be generated for consistency in the same way. Moreover, from what the authors state at lines 480-481, it looks like the confidence needs to be higher to generate a network with a larger number of interactions; this seems counterintuitive if the confidence refers to the minimal threshold for required interactions in the network (as I get from fig. 19 caption). Finally, the authors should mention how they perform the clustering of nodes to obtain the clusters shown in Fig. 20

- Two sentences at the end of the discussion section (line 552: the machinery of MLOS at least partially overlaps) and in the conclusions section (lines 677-678: mRNA processing and maintenance of the native structure of sequestered proteins in these organelles is provided by the same cellular machinery) are inconsistent with each other. I do thing the former one (in the discussion section) is more representative of the actual results shown by the authors

Minor comments

- line 40: IDRs -> IDPs (I guess)

- line 147: RBP is not defined

- Figure 1B: quadrants mentioned in the caption are not labeled in the plot

- line 275: to which paper does "their 2019 paper" refer to? 

- line 292: it is not clear to which protein set "these" refers to. All LLPS proteins (as from Figure 3) or just the ones with score > 0.9 (but which one? see the first major comment)

- caption of several figures: what are exactly LLPS-related proteins? I guess the LLPS class show in Fig. 3 and similar ones, but it should be stated explicitly.

- line 323: I do not understand the sentence "We have shown that the proportion of IDP and IDPRs in the proteome of acidosis-induced A-bodies is about 56%, which is 10% higher than the content of such proteins in heat-induced A-bodies". 56% (49+7 from Fig. 7A) should compare to 41% (39+2 from Fig. 3A)

- line 382: A-bodies have a ORDERED structure (not disordered as written by the authors), i.e. fibrillar, in comparison with nSBs

- line 411: Hsc70 should be changed to HspA8, I guess

In several places some words seem to be missing. I will give only a couple of examples (the likely missing word capitalized, when I can guess it) but there are several ones

- abstract, lines 28-29 "many of WHICH with high potential for spontaneous LLPS"

- introduction, lines 75-77 "the region between an ordered and a
disordered state characterized by largest a minimal impact on such systems can significantly change their properties and evolution over time" (in this case it is hard for me to find a way to make the sentence correct)

- introduction, lines 156-17 "spontaneous and phase separation that IS  induced by interactions with partners" (although here I would probably change in "phase separation that is both spontaneous and induced by interactions with partners")

- line 219,  a word is missing after "highly"

- line 222, singular person should be changed to plural

- line 621, "inter-network" -> internal network (I guess)

- I am not sure in which sense the authors use the word "Curiously" (a
couple of times). It sounds odd to me, I would change it maybe with noticeably (but I am not really sure what the authors mean).

- the wording "presenting identified" is used in the caption of several figures; it makes no sense to me

There are for sure other cases of sentences with English issues to be fixed

Author Response

Reviewer #2

Comments and Suggestions for Authors

The manuscript “Nucleolar stress-induced membrane-less organelles: Proteome analysis through the prism of liquid-liquid phase separation” by Mokin et al. analyzes datasets for the proteomes of nuclear membrane-less compartments. The focus is on Amyloid-bodies (A-bodies), generated upon heat shock or after exposure to low pH, and nuclear stress bodies (nSBs). The authors use publicly available datasets to perform intrinsic disorder analysis, STRING analysis, predict liquid-liquid phase separation (LLPS) and aggregation propensity. The aim is to gain new knowledge on the properties of A-bodies and nSBs.

Overall, this is valuable strategy that could be useful to define the properties of a wide variety of membrane-less compartments. In its present form, the text is sometimes difficult to understand. The manuscript needs to be read and corrected by a native English speaker. Importantly, the interpretation of the data analyses and the global conclusions drawn are not always solid.

GENERAL COMMENTS:

STRENGTHS. LLPS and membrane-less compartments are of prime interest to fundamental and applied research. A thorough characterization of the physicochemical properties of these compartments will be of value to different scientific fields. Having a defined set of analysis tools and a workflow that proposes how the analyses should be conducted will be useful.

WEAKNESSES. The analyses are restricted by the information available for the compartment proteomes. Given the difficulties with isolating membrane-less compartments (loss of proteins, contamination with components that do not reside in the compartment), the conclusions based on proteomics data are inherently prone to error.

The writing is verbose; many sentences are hard to follow, and the manuscript is riddled with typographical and grammatical mistakes. It is compulsory to fix these issues.

Reply. We are grateful to Reviewer 2 for carefully reading our work and providing valuable comments. We significantly improved the text quality. Of course, the quality of studied proteomes is the main limiting factor of our study. But significant differences between content of the proteins prone to spontaneous LLPS, induced LLPS and amyloid fibrils formation in the studied proteomes indirectly correlate with the fibril-like morphology of A-bodies and the liquid-like morphology of nSBs.

MAJOR POINTS:

[1]   The manuscript is often unclear; many statements need to be rephrased. It is mandatory that a native English speaker provides guidance for the revisions. In its current form, the manuscript has many errors that need to be corrected. Example, line 183: “Overall, this analysis reveal high level of disorder in the proteomes of these MLOs.”

Reply. We significantly improved the text quality.

[2]   Condense the Introduction. It is unfocused and lists a lot of details. The relevance of the excessive amount of information to the author’s study is not always clear. For instance, is it necessary to discuss details of rIGSRNA composition and properties?

Reply. We rewrote the Introduction. Since the rIGSRNA is considered as A-body scaffold,  we decided to include discussion pertaining to the rIGSRNA composition in the Introduction.

[3]   The authors have to discuss the limitations of their approach. The uncertainty of defining the proteome for membrane-less compartments is a major obstacle for the conclusions drawn by the authors.

Reply. Corrected

[4]   It will be helpful for the reader to get a clear idea about the workflow that is suitable for the characterization of membrane-less compartments. Please provide a flow chart that depicts (i) the recommended order of analyses (if applicable), (ii) the insight generated by a particular analysis, (iii) alternative approaches, and (iv) the limitation(s) of each type of analysis.

Reply. We added the appropriate section in the text.

[5]   Figure 17 is a disaster. The keys can hardly be read, and it is not clear what the figure is supposed to convey. Aside from being able to define the disorder for individual proteins, what is the take-home message?

Reply. Corrected

[6]   Rephrase: “... the morphology of A bodies and nSBs correlates with the protein composition of these organelles.”

This reviewer acknowledges that defining the factors that control the morphology of A-bodies and nSBs would be of interest. However, the authors do not at any point address the compartment “morphology” as it relates to form or size.

Reply. In our opinion, the content of the proteins prone to spontaneous LLPS, induced LLPS, and amyloid fibril formation in the studied proteomes indirectly correlates with the fibril-like morphology of A-bodies and the liquid-like morphology of nSBs.

[7]   Rephrase: “Comparative analysis of the proteomes of heat shock-induced and acidosis-induced A-bodies and nuclear stress-bodies made it possible to correlate the propensity of their proteins to phase separation with the morphological and functional properties of these organelles.”

Reply. Corrected

[8]   “The performed analysis allows us to conclude that mRNA processing and maintenance of the native structure of sequestered proteins in these organelles is provided by the same cellular machinery.”

        Without proper experimental evidence, this conclusion cannot be drawn. It is a blatant overinterpretation.

Reply. Corrected

MINOR POINTS:

[1]   NCI defines organelle as follows:” A small structure in a cell that is surrounded by a membrane and has a specific function.” This reviewer appreciates that the terminology used for compartments formed by LLPS is as best vague. Nevertheless, the term “organelle” should be avoided by the authors.

Reply. Corrected

[2]   Change the color scheme for Fig. 4, 5, 8, 9, 12, and 13. Using a larger variety of colors (i.e. beyond blues and grays), will make it easier to identify each category.

Reply. Corrected

[3]   Line 96: Define “etc.”. If details are not known, delete “etc.”.

Reply. Corrected

Reviewer 3 Report

The manuscript Nucleolar stress-induced membrane-less organelles: Proteome analysis through the prism of liquid-liquid phase separation” by Mokin et al. analyzes datasets for the proteomes of nuclear membrane-less compartments. The focus is on Amyloid-bodies (A-bodies), generated upon heat shock or after exposure to low pH, and nuclear stress bodies (nSBs). The authors use publicly available datasets to perform intrinsic disorder analysis, STRING analysis, predict liquid-liquid phase separation (LLPS) and aggregation propensity. The aim is to gain new knowledge on the properties of A-bodies and nSBs.

Overall, this is valuable strategy that could be useful to define the properties of a wide variety of membrane-less compartments. In its present form, the text is sometimes difficult to understand. The manuscript needs to be read and corrected by a native English speaker. Importantly, the interpretation of the data analyses and the global conclusions drawn are not always solid.

GENERAL COMMENTS:

Strengths. LLPS and membrane-less compartments are of prime interest to fundamental and applied research. A thorough characterization of the physicochemical properties of these compartments will be of value to different scientific fields. Having a defined set of analysis tools and a workflow that proposes how the analyses should be conducted will be useful.

Weaknesses. The analyses are restricted by the information available for the compartment proteomes. Given the difficulties with isolating membrane-less compartments (loss of proteins, contamination with components that do not reside in the compartment), the conclusions based on proteomics data are inherently prone to error.

The writing is verbose; many sentences are hard to follow, and the manuscript is riddled with typographical and grammatical mistakes. It is compulsory to fix these issues.

MAJOR POINTS:

[1]   The manuscript is often unclear; many statements need to be rephrased. It is mandatory that a native English speaker provides guidance for the revisions. In its current form, the manuscript has many errors that need to be corrected. Example, line 183: “Overall, this analysis reveal high level of disorder in the proteomes of these MLOs.”

[2]   Condense the Introduction. It is unfocused and lists a lot of details. The relevance of the excessive amount of information to the author’s study is not always clear. For instance, is it necessary to discuss details of rIGSRNA composition and properties?

[3]   The authors have to discuss the limitations of their approach. The uncertainty of defining the proteome for membrane-less compartments is a major obstacle for the conclusions drawn by the authors.

[4]   It will be helpful for the reader to get a clear idea about the workflow that is suitable for the characterization of membrane-less compartments. Please provide a flow chart that depicts (i) the recommended order of analyses (if applicable), (ii) the insight generated by a particular analysis, (iii) alternative approaches, and (iv) the limitation(s) of each type of analysis.

[5]   Figure 17 is a disaster. The keys can hardly be read, and it is not clear what the figure is supposed to convey. Aside from being able to define the disorder for individual proteins, what is the take-home message?

[6]   Rephrase: “... the morphology of A bodies and nSBs correlates with the protein composition of these organelles.”

This reviewer acknowledges that defining the factors that control the morphology of A-bodies and nSBs would be of interest. However, the authors do not at any point address the compartment “morphology” as it relates to form or size.

[7]   Rephrase: “Comparative analysis of the proteomes of heat shock-induced and acidosis-induced A-bodies and nuclear stress-bodies made it possible to correlate the propensity of their proteins to phase separation with the morphological and functional properties of these organelles.”

        This is an overstatement. What is the evidence that the “the propensity of their proteins to phase separation” can predict the compartment morphology, let alone “the functional properties”?

[8]   “The performed analysis allows us to conclude that mRNA processing and maintenance of the native structure of sequestered proteins in these organelles is provided by the same cellular machinery.”

        Without proper experimental evidence, this conclusion cannot be drawn. It is a blatant overinterpretation.

MINOR POINTS:

[1]   NCI defines organelle as follows:” A small structure in a cell that is surrounded by a membrane and has a specific function.” This reviewer appreciates that the terminology used for compartments formed by LLPS is as best vague. Nevertheless, the term “organelle” should be avoided by the authors.

[2]   Change the color scheme for Fig. 4, 5, 8, 9, 12, and 13. Using a larger variety of colors (i.e. beyond blues and grays), will make it easier to identify each category.

[3]   Line 96: Define “etc.”. If details are not known, delete “etc.”.

In its present form, the text is sometimes difficult to understand. The manuscript needs to be read and corrected by a native English speaker.

Author Response

Reviewer #3

Comments and Suggestions for Authors

IDP and IDR play important roles in the formation of MLOs. This manuscript presents a comprehensive analysis of the three proteomes of MLOs, including the heat shock-induced and acidosis-induced A-bodies and nSBs. The results reveal that these proteomes contain a higher percentage of disordered regions and proteins than the average. Further analysis demonstrates that the machinery of these MLOs shares biological processes while exhibiting different regulation processes. The results are presented clearly and in detail.

Regarding the manuscript, I recommend that Figures 18 and 19 display only one or two proteins, with the others included in the supplement. Additionally, Figure 19 should have a more detailed legend.

Reply. We are grateful to Reviewer 3 for carefully reading our work and providing valuable comments. We significantly improved the text quality.

There are several typos in the manuscript:

  1. Line 220: 'of proteins in the in the proteomes' should be corrected to 'of proteins in the proteomes.'

Reply. Corrected

  1. Line 222: 'that I expected' should be corrected to 'that are expected.

Reply. Corrected

  1. Line 519: 'Thereare 227' should be corrected to 'There are 227.'

Reply. Corrected

  1. Line 610: '10mhuman' should be corrected to '10 human.'

Reply. Corrected

Round 2

Reviewer 2 Report

The authors successfully replied to my comments.

However, some editing of the English editing is still required (at proof revision stage).

For example:

- lines 150-154: the same sentence is repeated twice

- lines 132-133:  "to undergo phase separate" -> "to undergo phase separation"

some editing of the English editing is still required (at proof revision stage).

For example:

- lines 150-154: the same sentence is repeated twice

- lines 132-133:  "to undergo phase separate" -> "to undergo phase separation"

Author Response

Comments and Suggestions for Authors

The authors successfully replied to my comments.

However, some editing of the English editing is still required (at proof revision stage).

For example:

- lines 150-154: the same sentence is repeated twice

- lines 132-133:  "to undergo phase separate" -> "to undergo phase separation"

RESPONSE: Thank you for pointing this out. Corresponding corrections were made in the revised manuscript.

Reviewer 3 Report

The revised manuscript “Nucleolar and nuclear stress-induced membrane-less organelles: A proteome analysis through the prism of liquid-liquid phase separation” by Mokin et al. analyzes datasets for the proteomes of nuclear membrane-less compartments. The proteome of Amyloid-bodies (A-bodies), generated upon heat shock or after exposure to low pH, and nuclear stress bodies (nSBs) are examined. The aim is to gain new knowledge on the properties of A-bodies and nSBs. To this end, the authors use publicly available datasets; they perform intrinsic disorder analysis, STRING analysis, predict liquid-liquid phase separation (LLPS) and aggregation propensities.

The revised manuscript has been improved compared with the previous version. However, several items still need to be addressed before publication.

GENERAL COMMENTS:

Strengths. LLPS and membrane-less compartments are of prime interest to fundamental and applied research in biology and the medical sciences. A thorough characterization of the physicochemical properties of these compartments will be of value to different scientific fields. Having a defined set of analysis tools and a workflow to conduct these analyses will be useful.

Weaknesses. The analyses are restricted by the information available for the compartment proteomes.

MAJOR POINTS:

[1]   The National Cancer Institute defines organelle as follows:” A small structure in a cell that is surrounded by a membrane and has a specific function.” The term “organelle” should be avoided by the authors when referring to non-membrane-bound compartments. That has to be corrected throughout the text.

[2]   Given the difficulties with isolating membrane-less compartments (loss of proteins, contamination with components that do not reside in the compartment), the conclusions based on proteomics data are inherently prone to error. The authors have to mention this in the Results section. Page 3, line 149 is a good place for such a statement.

[3]   For the analyses of proteins located in A-bodies and nSBs equal weight appears to be given to each individual protein present in the MLO.

This reviewer appreciates that it is difficult to factor in the copy numbers for each protein species located in an MLO. Nevertheless, this is a shortcoming that makes the conclusions regarding the overall properties of MLOs uncertain.

        The authors must mention this in the Discussion.

[4]   The following statement has to be toned down: “At the same time, according to our analysis, acidosis-induced A-bodies have more "liquid-droplet" properties.”

        A more appropriate statement would be: “At the same time, according to our analysis, acidosis-induced A-bodies have a larger number of protein species that promote "liquid-droplet" properties.”

[5]   Rephrase: “... the morphology of A bodies and nSBs correlates with the protein composition of these organelles.”

The authors do not at any point address the compartment “morphology” as it relates to form or size. This point was raised in the previous round of review. It must be addressed.

[6]   Rephrase: “Comparative analysis of the proteomes of heat shock-induced and acidosis-induced A-bodies and nuclear stress-bodies made it possible to indirectly correlate the propensity of their proteins to phase separation with the morphological properties of these organelles.”

        As mentioned in the first round of review, this is an overstatement that should be toned down:

        “Comparative analysis of the proteomes of heat shock-induced and acidosis-induced A-bodies and nSBs may suggest that the propensity of MLO proteins to phase separate impacts the morphological properties of these compartments. Future experiments will have to address this hypothesis directly in vitro or in vivo.

[7]   Page 13, line 356. The paper by Ninomiya et al. states for heat-shocked HeLa cells: ...”revealed that 141 proteins, most of which that have not yet been reported as nSB components, were specifically coprecipitated with HSATIII lncRNAs from the stressed cells”.

Explain where the number 133 is coming from in  “...133 proteins that are part of... ” are mentioned on page 13, line 356.

MINOR POINTS:

The revised version has been greatly improved. Nevertheless, it continues to have errors that need to be fixed. Please consider deleting in the manuscript the text that has been crossed out below.

[1]   Page 2, lines 62 and 63. Delete “Revolutionary”. “Ground-breaking” could be used instead.

[2]   Page 2, line 87. Change text to: ”...formed in the nucleus in response to stress.”

The original version: “formed in response to stress in the nucleus” sounds as if the stress was imposed specifically on the nucleus.

[3]   Page 3, line 110. Change text to: ...”another type of membrane-less compartments is formed in the nuclei of primate cells”.

[4]   Page 3, line 110. Should read: “foci of nSB formation”, not nSBs.

[5]   Page 4, line 150. “We analyzed the amino acid sequences of collected proteins for the potential propensity to spontaneous and induced liquid-liquid phase separation LLPS and the tendency of these proteins to gelation and amyloid fibrillation. Then we analyzed amino acid sequences of collected proteins for potential propensity to spontaneous and induced liquid-liquid phase separation and tendency of these proteins to gelation and amyloid fibrillation.”

        Delete the underlined sentence; it is a repetition.

[6]   Page 4, line 167 “The, proteomes...”. Delete comma.

[7]   Page 4, line 170. Replace “organelles” with “compartments”.

[8]   Page 5, line 183. “The quadrant that the protein is located determines its classification.” Change to: “The quadrant in which the protein is located determines its classification.”

[9]   Page 5, line 185. ...”predicted to be ordered to by CH-plot and disordered by CDF-plot.” Delete “to”.

[10] Page 6, line 190/191. “This classification is based on the accepted in the field practice to group proteins...”

[11] Page 7, line 251. Should read “...predisposition...”, not "predisposion”.

[12] Page 13, line 344/345. “...A-body proteomes represent proteins that are predisposed to inducible LLPS. This is significantly smaller than the content of such proteins in the heat shock-induced A-body proteome.”

[13] Page 15, line 380/381. “Distribution of protein intrinsic disorder, tendency to LLPS, roles in MLOs according to FuzDrop, and protein charge within the proteome ofnuclear stress bodies on.”

Correct to: “Distribution of protein intrinsic disorder, tendency to LLPS, roles in MLOs according to FuzDrop, and protein charge in the proteome of nSBs.”

[14] Page 16, line 394. “Apparently, it is rIGSRNAs molecules that...”, change to

“Apparently, rIGSRNA molecules that...”,

[15] Page 21, line 475. “Network was generated using medium confidence of 0.400 for minimal required interactions.” Change to: “The network was generated using medium confidence of 0.400 for minimal required interactions.”

The quality of English needs minor to moderate improvement. Some suggestions for improvement are included in the comments for the authors.

Author Response

Comments and Suggestions for Authors

The revised manuscript “Nucleolar and nuclear stress-induced membrane-less organelles: A proteome analysis through the prism of liquid-liquid phase separation” by Mokin et al. analyzes datasets for the proteomes of nuclear membrane-less compartments. The proteome of Amyloid-bodies (A-bodies), generated upon heat shock or after exposure to low pH, and nuclear stress bodies (nSBs) are examined. The aim is to gain new knowledge on the properties of A-bodies and nSBs. To this end, the authors use publicly available datasets; they perform intrinsic disorder analysis, STRING analysis, predict liquid-liquid phase separation (LLPS) and aggregation propensities.

The revised manuscript has been improved compared with the previous version. However, several items still need to be addressed before publication.

RESPONSE: We are thankful to this reviewer for careful reading of the manuscript and for the constructive criticism. We tried to address all the remaining concerns and revised manuscript accordingly

GENERAL COMMENTS:

STRENGTHS. LLPS and membrane-less compartments are of prime interest to fundamental and applied research in biology and the medical sciences. A thorough characterization of the physicochemical properties of these compartments will be of value to different scientific fields. Having a defined set of analysis tools and a workflow to conduct these analyses will be useful.

RESPONSE: We glad to see that you have found out research and assembled set of tools useful.

WEAKNESSES. The analyses are restricted by the information available for the compartment proteomes.

RESPONSE: Thank you for pointing this out. We do understand that the reported analyses are restricted by the information available for the compartment proteomes. However, in our view, such limitation is naturally present in scientific research, and even the most comprehensive study is limited by the available information.

MAJOR POINTS:

[1]   The National Cancer Institute defines organelle as follows:” A small structure in a cell that is surrounded by a membrane and has a specific function.” The term “organelle” should be avoided by the authors when referring to non-membrane-bound compartments. That has to be corrected throughout the text.

RESPONSE: We respectfully disagree with this request. In fact, most of the papers dedicated to the biomolecular condensates are using “membrane-less organelle” or “membraneless organelle” or “non-membrane bound organelle” terms for the description of these entities. According to our Web of Science analysis, only 8 papers used “non-membrane bound compartment" term, whereas “membrane-less organelle”, “membraneless organelle”, and “non-membrane bound organelle” were mentioned in 258, 737, and 32 papers, respectively. Therefore, we decided to use the “membrane-less organelle” throughout this manuscript.

[2]   Given the difficulties with isolating membrane-less compartments (loss of proteins, contamination with components that do not reside in the compartment), the conclusions based on proteomics data are inherently prone to error. The authors have to mention this in the Results section. Page 3, line 149 is a good place for such a statement.

RESPONSE: Thank you for pointing this out. The corresponding statement was added to the revised manuscript. Page 3-4, lines 148-151: “Furthermore, one should also keep in mind that given the difficulties associated with isolating membrane-less organelles (loss of proteins, contamination with components that do not reside in the compartment), the conclusions based on the proteomics data are inherently prone to error.”

[3]   For the analyses of proteins located in A-bodies and nSBs equal weight appears to be given to each individual protein present in the MLO.

This reviewer appreciates that it is difficult to factor in the copy numbers for each protein species located in an MLO. Nevertheless, this is a shortcoming that makes the conclusions regarding the overall properties of MLOs uncertain.

        The authors must mention this in the Discussion.

RESPONSE: Thank you for pointing this out. The corresponding statement was added to the revised manuscript. Page 23, lines 266-272: “Finally, we have to emphasize that one of the shortcomings of this study is based on the fact that in our analyses of proteins located in A-bodies and nSBs, equal weights were given to each individual protein present in the MLO. However, it is possible that levels of different proteins located within MLOs could be different, with some of them being more abundant than others. Since it is difficult to factor in the copy numbers for each protein species in an MLO, this represents a shortcoming that makes the conclusions regarding the overall properties of MLOs less certain.”

[4]   The following statement has to be toned down: “At the same time, according to our analysis, acidosis-induced A-bodies have more "liquid-droplet" properties.”

        A more appropriate statement would be: “At the same time, according to our analysis, acidosis-induced A-bodies have a larger number of protein species that promote "liquid-droplet" properties.”

RESPONSE: Thank you for pointing this out. This statement was amended as recommended.

[5]   Rephrase: “... the morphology of A bodies and nSBs correlates with the protein composition of these organelles.”

The authors do not at any point address the compartment “morphology” as it relates to form or size. This point was raised in the previous round of review. It must be addressed.

RESPONSE: Thank you for pointing this out. To address this issue, the corresponding paragraph was changed to read: “A comparative analysis of the proteomes of A-bodies formed in response to heat shock, acidosis, and nuclear stress bodies showed that the morphology and mechanical properties of A bodies and nSBs correlates with the protein composition of these organelles. A-bodies are MLOs that transform into fibrillar structures during their biogenesis, contain significantly fewer proteins prone to LLPS and have less highly disordered proteins compared to nSBs, which normally have liquid-droplet characteristics (Table 1).”

[6]   Rephrase: “Comparative analysis of the proteomes of heat shock-induced and acidosis-induced A-bodies and nuclear stress-bodies made it possible to indirectly correlate the propensity of their proteins to phase separation with the morphological properties of these organelles.”

        As mentioned in the first round of review, this is an overstatement that should be toned down:

        “Comparative analysis of the proteomes of heat shock-induced and acidosis-induced A-bodies and nSBs may suggest that the propensity of MLO proteins to phase separate impacts the morphological properties of these compartments. Future experiments will have to address this hypothesis directly in vitro or in vivo.”

RESPONSE: Thank you for pointing this out. This statement was amended as recommended.

[7]   Page 13, line 356. The paper by Ninomiya et al. states for heat-shocked HeLa cells: ...”revealed that 141 proteins, most of which that have not yet been reported as nSB components, were specifically coprecipitated with HSATIII lncRNAs from the stressed cells”.

Explain where the number 133 is coming from in  “...133 proteins that are part of... ” are mentioned on page 13, line 356.

RESPONSE: Thank you for pointing this out. Please find below explanation of why we are studying 133 and not 141 proteins. In the paper by Ninomiya et al., data on the proteins included in the nSB are presented in the form of a table of proteins associated with HSATIII detected by the ChIRP-MS method. The methodology of the original article does not explicitly indicate which of the proteins in the table are classified by the authors as part of the nSB. However, the data in the table are sorted in descending order of the Score ("Score: (the mean number of peptides (42-HSATIII) + 0.00001)/( the mean number of peptides (37-HSATIII) + 0.00001)"). Based on the data in that table, proteins up to the line 141 have ?????≥ 33333.3. We assumed that this is the boundary value, below which proteins are not considered to be part of the nSB. Intuitively, it seems that this means that a protein is considered to be a part of the nSB if it interacted with HSATIII in at least 2 out of 3 experiments under heat stress conditions and never did so under normal conditions. One should also keep in mind that the aforementioned table contains both, protein complexes or proteins of the same family (written with |, for example RBMX|RBMXL1|RBMXL2|RBMXL3), and individual proteins (for example, separately RBMXL1). Therefore, some of the entries in the table have duplicate Protein IDs. Since only we can only analyze individual proteins with the non-redundant IDs, such duplicates were removed. As a result, we got a set of 131 unique Protein IDs. We have also added two nSB marker proteins to this list, HSF1 and HSF2. This gave us a total of 133 proteins for analysis. The corresponding clarification is added to the revised manuscript,

MINOR POINTS:

The revised version has been greatly improved. Nevertheless, it continues to have errors that need to be fixed. Please consider deleting in the manuscript the text that has been crossed out below.

[1]   Page 2, lines 62 and 63. Delete “Revolutionary”. “Ground-breaking” could be used instead.

[2]   Page 2, line 87. Change text to: ”...formed in the nucleus in response to stress.”

The original version: “formed in response to stress in the nucleus” sounds as if the stress was imposed specifically on the nucleus.

[3]   Page 3, line 110. Change text to: ...”another type of membrane-less compartments is formed in the nuclei of primate cells”.

[4]   Page 3, line 110. Should read: “foci of nSB formation”, not nSBs.

[5]   Page 4, line 150. “We analyzed the amino acid sequences of collected proteins for the potential propensity to spontaneous and induced liquid-liquid phase separation LLPS and the tendency of these proteins to gelation and amyloid fibrillation. Then we analyzed amino acid sequences of collected proteins for potential propensity to spontaneous and induced liquid-liquid phase separation and tendency of these proteins to gelation and amyloid fibrillation.”

        Delete the underlined sentence; it is a repetition.

[6]   Page 4, line 167 “The, proteomes...”. Delete comma.

[7]   Page 4, line 170. Replace “organelles” with “compartments”.

[8]   Page 5, line 183. “The quadrant that the protein is located determines its classification.” Change to: “The quadrant in which the protein is located determines its classification.”

[9]   Page 5, line 185. ...”predicted to be ordered to by CH-plot and disordered by CDF-plot.” Delete “to”.

[10] Page 6, line 190/191. “This classification is based on the accepted in the field practice to group proteins...”

[11] Page 7, line 251. Should read “...predisposition...”, not "predisposion”.

[12] Page 13, line 344/345. “...A-body proteomes represent proteins that are predisposed to inducible LLPS. This is significantly smaller than the content of such proteins in the heat shock-induced A-body proteome.”

[13] Page 15, line 380/381. “Distribution of protein intrinsic disorder, tendency to LLPS, roles in MLOs according to FuzDrop, and protein charge within the proteome ofnuclear stress bodies on.”

Correct to: “Distribution of protein intrinsic disorder, tendency to LLPS, roles in MLOs according to FuzDrop, and protein charge in the proteome of nSBs.”

[14] Page 16, line 394. “Apparently, it is rIGSRNAs molecules that...”, change to

“Apparently, rIGSRNA molecules that...”,

[15] Page 21, line 475. “Network was generated using medium confidence of 0.400 for minimal required interactions.” Change to: “The network was generated using medium confidence of 0.400 for minimal required interactions.”

RESPONSE: Thank you for pointing this out. All the requested changes were introduced to the revised manuscript.